**Investigation**

# SoxB1-mediated chromatin remodeling promotes sensory neuron differentiation in planarians

Mallory L. Cathell,[1] Mohamad A. Auwal,[1] Sarai Alvarez Zepeda,[1] Kelly G. Ross (ID),[1] Ricardo M. Zayas (ID)[1,*]

[1]Department of Biology, San Diego State University, 5500 Campanile Dr., San Diego, CA 92182-4614, United States

*Corresponding author: Email: rzayas@sdsu.edu

Understanding how adult stem cells generate neurons is critical for advancing regenerative medicine. However, few in vivo models enable studying how stem cell fates are specified as neurons in an adult body. The planarian *Schmidtea mediterranea* provides a powerful system for investigating these mechanisms, owing to its abundant adult pluripotent stem cells, termed neoblasts, and its capacity to regenerate a molecularly complex nervous system. The SoxB1 family of transcription factors is broadly implicated in ectodermal lineage commitment. In planarians, the SoxB1 homolog *soxB1-2* has been shown to promote neural and epidermal differentiation. However, the mechanisms by which *soxB1-2* influences chromatin dynamics and transcriptional programs during adult neurogenesis remain unknown. To address this, we performed ATAC-seq and RNA-seq on neural-rich head tissues to assess how *soxB1-2* RNAi knockdown alters chromatin accessibility and gene expression. Disrupting *soxB1-2* resulted in reduced chromatin accessibility and transcriptional downregulation at neural and epidermal loci, consistent with a pioneer-like role in chromatin priming. We identified 31 candidate downstream targets with concordant accessibility and expression changes, including the transcription factors *castor* and *mecom*, which regulate mechanosensory and ion transport genes. Head tissue sampling enabled the detection of *soxB1-2*-responsive genes within rare neural subtypes that were missed in our previous whole-body RNA-seq experiments. These findings offer mechanistic insight into adult ectodermal lineage specification and establish a framework for understanding chromatin-mediated neurogenesis in regenerative systems.

Keywords: ATAC-seq; epigenetics; *mecom*; *castor*; stem cells

## Summary

Adult stem cells can regenerate neurons in some animals, but the mechanisms that control this process remain unclear. This study examined how the transcription factor SoxB1-2 regulates neuron formation in the regenerative flatworm *Schmidtea mediterranea* by combining chromatin accessibility profiling (ATAC-seq) and gene expression analysis (RNA-seq) following reduced SoxB1-2 activity. Loss of SoxB1-2 resulted in a decrease in both chromatin accessibility and expression of neural and sensory genes, including neural transcription factors such as *mecom* and *castor*. These results support a model in which SoxB1-2 promotes sensory neuron differentiation through chromatin accessibility changes at sensory neuron gene loci. Together, these findings link SoxB1-2 to chromatin accessibility and sensory neuron gene regulation during adult neurogenesis in planarians.

## Introduction

The nervous system is necessary for most of the physiological and behavioral activities of metazoans. However, many adult organisms lack the ability to regenerate lost neurons de novo. Understanding the molecular programming of stem cells during adult neurogenesis may inform strategies to induce neural fates and restore damaged or lost neurons. One highly conserved class of transcription factors that has roles in neural programming during development includes the SoxB1 group proteins. Diversification of the SoxB family into SoxB1 and SoxB2 subfamilies occurred prior to vertebrate diversification, and the ancestral bilaterian had both a SoxB1 and SoxB2 gene (Zhong et al. 2011). Thus, members of the SoxB1 family are conserved across bilaterians, including SoxB1 family members Sox1, 2, and 3 in mammals, of which Sox2 is critical for maintaining stem cell potency and directing the specification of multiple neural progenitor lineages (Zhang and Cui 2014). Beyond development, SoxB1 genes are expressed in adult tissues and contribute to regenerative processes across diverse organisms (Kamachi and Kondoh 2013; Reiprich and Wegner 2015; She and Yang 2015).

SoxB1 homologs regulate neurogenesis across various neural lineages by modulating transcription factor networks that drive terminal fate decisions (Miyagi et al. 2009). They are thought to function as pioneer transcription factors, capable of binding to closed chromatin and altering accessibility, thus facilitating the binding of downstream factors and initiating broad regulatory cascades (Stevanovic et al. 2021). Despite their well-characterized roles in development, the genomic activity and function of SoxB1 genes influencing cell fate specification and differentiation during adult neural regeneration remain poorly understood.

Integrating epigenomic and transcriptomic approaches offers a path to dissect how SoxB1 genes orchestrate adult stem cell-driven neurogenesis. This approach enables identification of SoxB1 regions targeted for chromatin alteration and links the

changes in chromatin dynamics to gene regulation, offering mechanistic insight into SoxB1-driven cell fate specification. One such strategy involves comparing chromatin accessibility and gene expression profiles in systems where SoxB1 activity is perturbed to controls. Characterization of these target loci can illuminate how chromatin-level regulation underlies SoxB1-mediated neurogenesis.

The planarian *Schmidtea mediterranea* presents an ideal model for investigating neural regeneration. *Schmidtea mediterranea* harbors a large population of pluripotent stem cells, enabling complete tissue regeneration, including a molecularly complex nervous system (Cebrià 2007; Rink 2013, 2018; Ross et al. 2017). Previous work has shown that inhibition of the planarian SoxB1 gene, *soxB1-2*, results in severe phenotypes, including seizure-like behaviors and loss of mechanosensation (Ross et al. 2018). Co-expression analyses, conducted via in situ hybridization with cell-type-specific markers, along with pseudotime analysis of single-cell RNA sequencing data, suggested that *soxB1-2*[+] stem cells constitute ectodermal progenitors destined for epidermal or neuronal fates (Ross et al. 2018).

Functional screens from our lab have identified *soxB1-2* downstream targets, including the BRN3/POU4 homolog *pou4-2*, which plays a role in specifying a subset of mechanosensory neurons (McCubbin et al. 2025). Knockdown of *soxB1-2* leads to depletion of the *pou4-2*[+] mechanosensory population, suggesting a positive regulatory relationship (McCubbin et al. 2025). Additionally, RNAi of *pou4-2* results in loss of the sensory neuron terminal gene *pkd1L-2*, a phenotype also observed following *soxB1-2* disruption (Ross et al. 2018; McCubbin et al. 2025). However, the mechanisms by which *soxB1-2* regulates sensory neural fate trajectories remain unclear. One hypothesis is that *soxB1-2* acts as a pioneer transcription factor in planarians, binding to silent chromatin and modifying the chromatin landscape at target loci to permit binding of additional transcription factors that direct the differentiation of terminal sensory and epidermal lineages.

In this study, we tested the hypothesis that *soxB1-2* primes chromatin for the acquisition of sensory and epidermal progenitor fates through the integration of epigenetic and transcriptomic approaches focused on isolated planarian head tissues to enrich for neuronal cell populations. We assessed changes in chromatin accessibility (via ATAC-seq) and gene expression (via RNA-seq) following *soxB1-2* RNAi knockdown to identify direct and functional targets. Our findings revealed that loss of *soxB1-2* leads to reduced chromatin accessibility at genomic regions associated with sensory neuronal and epidermal genes, some of which encode transcription factors. Among these, we identified *mecom* and *castor* transcription factor genes, which we found are involved in the regulation of mechanosensory signaling genes. Together, these results provide mechanistic evidence that *soxB1-2* functions as a pioneer factor in priming chromatin for the emergence of sensory neuronal and epidermal progenitor fates.

## Materials and methods
### Animal husbandry
Asexual *S. mediterranea* (clonal strain CIW4) planarians were kept in 9-cup Ziploc reusable containers in 1× Montjuïc salts (1.6 mM NaCl, 1.0 mM CaCl$_2$, 1.0 mM MgSO$_4$, 0.1 mM MgCl$_2$, 0.1 mM KCl, 1.2 mM NaHCO$_3$) dissolved in ultrapure filtered water at 20 °C. Animals were fed beef calf liver weekly, as previously described (Merryman et al. 2018). Animals were starved for 1 week prior to all experiments.

### RNA interference
Double-stranded RNA (dsRNA) was expressed in HT115 *Escherichia coli* containing cDNA inserts in either pJC.53.2 (Collins et al. 2010) or pPR-T4P (Liu et al. 2013) as described previously (Ross et al. 2018). Worms measuring 3–4 mm (for in situ hybridization) or 5–7 mm (for ATAC-seq and RNA-seq) were fed dsRNA, mixed 1:3 with a paste composed of 200 μL ultrapure water, 500 μL pureed calf liver, and 40 μL food dye. Animals were cleaned with 1× Montjuïc salt exchanges after each feeding and again the following day. Animals were fed dsRNA twice per week, for a total of five to ten feedings depending on the target gene. *soxB1-2* RNAi animals received five feedings, *mecom* RNAi animals received six feedings, and *castor* RNAi animals received 10 feedings. *Green fluorescent protein* (*gfp*) dsRNA was used as the negative control. For *soxB1-2* RNAi experiments, animals were amputated anterior to the pharynx 3 days after the fifth feeding for RNA-seq and ATAC-seq or fixed for in situ hybridization 7 days after the fifth feeding. For *mecom* and *castor* RNAi experiments, whole worms (3–4 mm) were fixed for in situ hybridization or processed for RNA-seq 7 days after the sixth feeding or tenth feeding, respectively.

### ATAC-seq assay and analysis
Two independent replicates of both *soxB1-2* RNAi and *gfp* control groups (30 heads per replicate) were placed directly into ATAC-seq lysis buffer (10 mM Tris pH 7.4, 10 mM NaCl, 3 mM MgCl$_2$, 0.1% Digitonin, 0.1% Tween-20, 0.1% NP-40, in water) and pipetted up and down until homogeneous. The slurry was run through a 50-micron filter, and the ATAC-seq assay was performed as previously described (Corces et al. 2017) with the Illumina tagment DNA enzyme and buffer kit (cat# 20034197). Samples were purified using Zymo DNA Clean and Concentrator-5 Kit (cat# D4014) and stored at −20 °C until sequencing. Samples were quality-checked on an Agilent 2100 Bioanalyzer and sequenced to a read depth of 38 million paired-end reads on the Illumina NovaSeq 6000 at the University of California, San Diego IGM Genomics Center, Inc. (San Diego, CA).

ATAC-seq libraries were assessed for quality using FastQC (v0.11.9) (Andrews 2010), and fragment size distributions were evaluated with ATACseqQC (v1.26.0) (Ou et al. 2018). Adapter trimming and merging of forward and reverse reads were performed using NGmerge (v0.3) (Gaspar 2018). Trimmed reads were aligned to both the S3h1 and S2F2 *S. mediterranea* genome assemblies using Bowtie2 (v2.4.4) (Langdon 2015). Samtools (v1.2.0) software was used to remove mitochondrial and improperly paired reads and to generate BAM files (Li et al. 2009). To control for differences in sequencing depth across ATAC-seq libraries, reads were down-sampled prior to peak calling and footprinting to equalize the number of properly paired, uniquely mapped reads between conditions. Subsampling was performed using samtools (v1.17) with the -s flag, which performs Bernoulli sampling on alignments. For each replicate, BAM files were name-sorted and subsampled using a unique random seed and fraction to achieve matched read counts across samples. Picard (v2.26.4) was used to remove PCR duplicates. Deduplicated BAM files were used for peak calling with Genrich (BioGrids) using parameters -j -v -p 0.01 -a 150. Peaks were called for individual and pooled replicates. Genrich-generated BedGraph files were converted to BroadPeak format and processed as previously described (Reske et al. 2020). Consensus peaks were mapped to gene annotation models for both genomes using BEDTools (v2.31.1) (Quinlan and Hall 2010). RPKM-normalized coverage tracks (bigWig) and profile plots

were generated using deepTools (v3.5.1) (Ramirez et al. 2014). Differential accessibility analysis between *gfp* controls and *soxB1-2* RNAi samples was performed using csaw (v1.36.0) (Lun and Smyth 2016), applying an FDR cutoff ≤ 0.05 as described previously (Reske et al. 2020). Gene Ontology enrichment was performed using enrichGO from clusterProfiler (v4.12.6) (Yu et al. 2012). The resulting datasets were compared to the human proteome using BLASTX (cut-off e-value < $1e^{-3}$). Human UniProt IDs (UniProt Consortium 2015) were used as input for annotation and overrepresentation analysis. Statistical analyses were carried out in R (v4.4.0) (R Core Team 2024), and figures were generated using ggplot2 (Wickham 2009).

## Motif enrichment analysis

Differentially accessible (DA) regions were first identified independently using csaw and edgeR (Bioconductor v3.19), based on a unified peak set generated across all ATAC-seq samples. Peaks with FDR ≤ 0.05 were considered significantly differentially accessible. Peaks with positive log₂FC values were classified as more accessible in *soxB1-2(RNAi)* samples, while peaks with negative log₂FC values were classified as more accessible in controls. Significant DA peaks were separated by direction and resized to a fixed width of 350 bp centered on the peak midpoint using GenomicRanges to standardize regions for footprinting. Footprint scores were computed across the unified peak set for each biological replicate using TOBIAS FootprintScores (Bentsen et al. 2020). Condition-level footprint signal tracks were generated by averaging replicate footprint bigWig files using bigwigCompare (deepTools 3.5.6; –operation mean) to produce a single representative footprint track for control and *soxB1-2* RNAi samples.

Differential TF binding analysis was performed using TOBIAS BINDetect (Bentsen et al. 2020), supplying the averaged footprint tracks, the unified resized peak set, and the S2F2 *S. mediterranea* genome. A curated set of transcription factor motifs derived from JASPAR (Castro-Mondragon et al. 2022) was used as input motifs. BINDetect was run with default settings to identify motif sites and infer differential motif occupancy between conditions (Quinlan and Hall 2010; Ramirez et al. 2014; Bentsen et al. 2020).

## Gene identification and cloning

The *soxB1-2* cDNA used in all experiments was previously cloned into pJC.53.2 (Collins et al. 2010; Ross et al. 2018) in our lab. All other gene sequences were identified through PlanMine (Rozanski et al. 2019) using the dd_Smed_v6 transcriptome and cloned from *S. mediterranea* cDNA into pPR-T4P (Liu et al. 2013) using ligation-independent cloning and gene-specific primers as described in Adler and Alvarado (2018) (all primer sequences and gene identifiers listed in Supplementary Table 6). In the text, dd_Smed_v6 transcripts are referred to as *dd_transcript no.* for brevity.

## Wholemount in situ hybridization

Antisense probes were generated from DNA templates amplified from pJC.53.2 (Collins et al. 2010; Ross et al. 2018) or pPR-T4P (Liu et al. 2013), incorporating digoxigenin-labeled UTPs, as previously described (Pearson et al. 2009). Samples were incubated in 5% N-acetyl cysteine (NAC) in 1✕ PBS for five minutes to euthanize the animals, then fixed in 4% formaldehyde in 1✕ PBS with 0.1% Triton X-100 for 25 min. Animals were processed for in situ hybridization as described previously (King and Newmark 2013), except for *mecom*, a low-abundance transcript, for which hybridization and heated washes were performed at 58 °C.

## RNA-seq assay and analysis

For *soxB1-2* RNAi experiments, head fragments from four independent *gfp* control and six independent *soxB1-2* RNAi replicates were homogenized in TRIzol (Invitrogen), and RNA was extracted and purified according to the manufacturer's protocol. For *mecom* and *castor* RNAi experiments, the same procedure was applied to whole worms, using five independent *gfp* controls and five *mecom* or castor RNAi replicates. RNA was treated with the TURBO DNA-free Kit and column-purified using the Zymoclean Miniprep Kit (cat. no. 11-330). RNA libraries were sequenced on an Illumina NovaSeq 6000 platform to a depth of 24 million reads per sample (12 million paired-end reads) at MedGenome, Inc. (Foster City, CA). Reads were pseudo-aligned to the S2F2 transcriptome using kallisto (Bray et al. 2016), and differential gene expression analysis was performed using Bioconductor (Huber et al. 2015) and DESeq2 (Love et al. 2014) with an FDR (*P*-adj) cut-off value ≤ 0.1 and fold-change threshold of 1.4. Gene Ontology enrichment was conducted using enrichGO from clusterProfiler (v4.12.6) (Yu et al. 2012). Gene sets were compared to the human proteome using BLASTX (e-value cut-off < $1e^{-3}$). Resulting UniProt human gene IDs were used for annotation and GO overrepresentation analysis. Statistical analyses were carried out in R (v4.4.0) (R Core Team 2024), and figures were generated using ggplot2 (Wickham 2009).

## Results
### Neural genes exhibit reduced chromatin accessibility following *soxB1-2* RNAi

To investigate how *soxB1-2* activity influences chromatin accessibility, we performed ATAC-seq (Buenrostro et al. 2015) on head tissues (amputated in the "neck" region anterior to the pharynx and posterior to the auricles; see Supplementary Fig. 1) from *soxB1-2(RNAi)* and *gfp(RNAi)* (control) animals to enrich for neural cell populations. The optimal RNAi feeding regimen and time-point for obtaining *soxB1-2* knockdown samples were determined based on prior studies conducted in our laboratory (Ross et al. 2018). Sequencing data were aligned to the *S. mediterranea* genome assembly (S3h1) (Ivankovic et al. 2024), and peaks were mapped to the nearest annotated gene model (see Materials and Methods). Peak calling identified 27,560 reproducible accessible chromatin regions in control animals and 26,679 regions in *soxB1-2(RNAi)* animals with 66 and 64% overlap between biological replicates, respectively (*P* ≤ 0.01, minimum peak length of 150 bp; see fragment size distributions from each ATAC-seq replicate in Supplementary Fig. 2a). Replicate accessibility profiles were highly correlated (Pearson r = 0.96 for both conditions; Supplementary Fig. 2b, c and Supplementary Table 1), confirming strong reproducibility across samples.

Subsequent differential accessibility analysis using csaw (FDR ≤ 0.05; see Materials and Methods) identified 264 significantly differentially accessible (DA) peaks. The majority of these (257 peaks) were present in controls but significantly reduced in *soxB1-2(RNAi)* animals (down). In comparison, only seven peaks were significantly increased in *soxB1-2(RNAi)* animals relative to controls (up) (Supplementary Table 1). This overall reduction in accessible peaks in the *soxB1-2(RNAi)* animals was also observed as a cumulative decrease in signal across all gene bodies and promoter regions in the *soxB1-2* RNAi groups (Fig. 1a–d). Thirteen genes associated with transcription factor binding were identified within DA chromatin regions, such as *castor* (Fig. 1d, e), which has been linked to neurodevelopmental processes in both invertebrate and vertebrate models (Isshiki et al. 2001). Most of the peaks

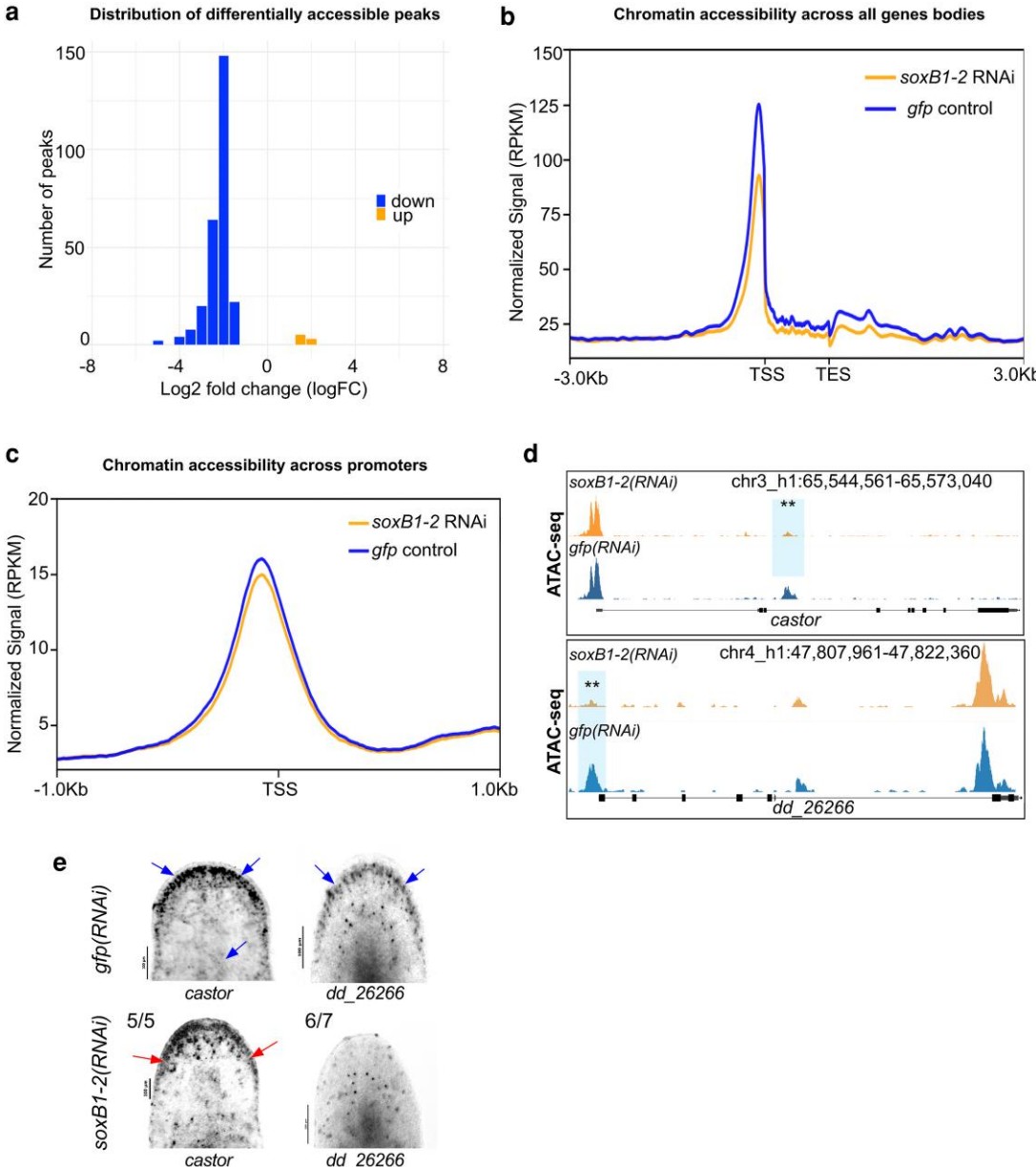

**Fig. 1.** *soxB1-2* RNAi reduces chromatin accessibility in *S. mediterranea*. a) Histogram of log$_2$ fold changes (log$_2$FC) in differentially accessible ATAC-seq peaks between *gfp* and *soxB1-2* RNAi groups, with 264 peaks exhibiting decreased accessibility (blue) and seven peaks showing increased accessibility (orange) (FDR ≤ 0.05). Sample size: N = 2 for both control and *soxB1-2* RNAi groups. b) Profile plot of global ATAC-seq signal across gene bodies in *gfp* control (blue) and *soxB1-2* RNAi (orange) groups. TSS, transcription start site; TES, transcription end site. c) Profile plot of global ATAC-seq signal across promoter regions in *gfp* control (blue) and *soxB1-2* RNAi (orange) groups. TSS, transcription start site. d) Genome tracks showing reduced chromatin accessibility at the *mecom* and dd_26266 locus. e) In situ hybridization of genes enriched in neural and ciliated neural cell populations that display reduced chromatin accessibility post-*soxB1-2* RNAi. Blue arrows indicate regions of in situ signal loss following *soxB1-2* RNAi, with specific anatomical areas marked using red arrows. Sample size: For controls, N ≥ 3 worms were analyzed, and all samples showed consistent expression patterns. For RNAi-treated animals, the number of biological replicates (N) is indicated at the top left of each treatment group.

that were shared between the *gfp* and *soxB1-2* RNAi groups were located in distal intergenic regions (54.7 and 57.54%, respectively; Supplementary Fig. 3). Promoter regions accounted for 22% of peaks in controls and 21.25% in *soxB1-2* RNAi samples, while internal intronic regions contained 17.53 and 16.23%, respectively. Only a small fraction mapped to internal exonic regions, 3% in controls and 2.56% in *soxB1-2* RNAi. In contrast, DA peaks showed a markedly different distribution: only 25.52% were in distal intergenic regions, while 36.39% mapped to internal intronic regions and 9.96% to internal exonic regions (Supplementary Fig. 3).

Distinguishing direct chromatin-opening activity of SoxB1-2 from indirect effects mediated through downstream regulators is an important consideration. To evaluate this, we performed motif enrichment analysis following Neiro et al. (2022) using differentially accessible regions between control and *soxB1-2(RNAi)* animals. Sox-family binding motifs, including SoxB1-like sites, were significantly enriched in regions that remain accessible in control animals relative to *soxB1-2(RNAi)* (Supplementary Fig. 4). This supports a model in which SoxB1-2 contributes to establishing or maintaining chromatin accessibility at these loci,

consistent with pioneer-like activity as described for other SoxB1 factors (Neiro et al. 2022).

To validate ATAC-seq results and determine whether genes that lose chromatin accessibility are downstream targets of *soxB1-2*, we selected two genes, *castor* and *dd_26266*, for in situ hybridization following *soxB1-2* RNAi. These genes are expressed in the same neural or ciliated neural clusters as *soxB1-2* (Fincher et al. 2018). In situ analysis revealed reduced expression of *castor* and *dd_26266* in the head margin (Fig. 1e), a region enriched in sensory neurons (Okamoto et al. 2005; Wang et al. 2016; Reho et al. 2022). Additionally, *castor* exhibited further expression loss in the dorsal epidermis and dorsal ciliated stripe (an organ associated with water flow sensing in planarians) (Hyman 1951; Farnesi and Tei 1980; Ross et al. 2018) (Fig. 1e). The loss of chromatin accessibility and gene expression in sensory neuron-rich regions supports a key role of *soxB1-2* in regulating downstream genes with potential roles in neuronal specification or terminal differentiation, including other transcription factors.

## *soxB1-2* RNAi results in loss of sensory and neural gene expression

To examine the relationship between gene regulation and chromatin accessibility in the *soxB1-2* RNAi amputation scheme, we repeated *soxB1-2* RNAi experiments and performed RNA-seq on RNA extracted from animal heads (Supplementary Fig. 1). RNA-seq analysis identified 533 downregulated genes meeting our significance cut-offs (fold-change $\geq$ 1.4, $P$-adj $\leq$ 0.1) (Fig. 2a, Supplementary Table 2). Among these, we observed a significant reduction in genes previously shown to be downstream targets of *soxB1-2* that are involved in neural and mechanosensory function. These included *pou4-2, pkd1L-1, pkd1L-2, pkd1L-3, pkd2L-1, pkd2L-2, pkd2-1,* and *pkd2-4* (Fig. 2a) (Ross et al. 2024; McCubbin et al. 2025).

To validate the differential expression results, we isolated all genes that were significantly differentially expressed between *gfp* controls and *soxB1-2* RNAi-treated animals and generated a heatmap (Supplementary Fig. 5). This visualization confirmed consistent expression patterns across replicates, with upregulated genes in the control group showing corresponding downregulation in the RNAi group, and vice versa, indicating robust sample clustering for this gene set. To further investigate these expression changes and validate patterns observed in our dataset, we selected *no mechanoreceptor protein channel c* (*nompc*), *dd_17839, formimidoyltransferase cyclodeaminase* (*ftcd*), *ms310138,* and *receptor-type tyrosine-protein phosphatase S* (*ptprs*) as representative genes downregulated in the *soxB1-2(RNAi)* group. These genes were co-expressed with *soxB1-2* specifically within neural and ciliated neural populations in published single-cell RNA-seq datasets (Wurtzel et al. 2015) (Supplementary Fig. 6). To assess whether they were potential downstream targets of *soxB1-2*, we performed in situ hybridization on *gfp* and *soxB1-2* RNAi animals. Consistent with the RNA-seq results, these genes exhibited reduced expression following *soxB1-2* RNAi, particularly in sensory neuron-enriched regions concentrated in the head rim and dorsal ciliated stripe (Fig. 2b).

The observed transcriptional changes following *soxB1-2* RNAi prompted us to examine how these patterns compared to those in previous RNA-seq datasets. Given that our prior studies were conducted on RNA extracted from whole worms (Ross et al. 2018), we sought to determine whether broader tissue sampling may have masked gene expression changes in rare neural populations in the whole-worm datasets. Comparing this head fragment dataset to previous whole-worm *soxB1-2* RNAi RNA-seq results identified 134 shared downregulated genes (Supplementary Table 2). Gene Ontology (GO) enrichment analysis indicated that shared genes are primarily associated with "ion homeostasis", "transmembrane transport regulation", and "cell communication", with key processes including "negative regulation of ion and cation transport", "sodium ion transmembrane transport", and "response to metal ions" (Supplementary Fig. 7a). Additionally, the presence of electrical coupling-related terms suggests a role in direct cell-to-cell communication, particularly in excitable tissues such as neurons and muscle.

In contrast, 396 genes were uniquely identified in the RNA-seq dataset from head tissue compared to our previously published day-14 whole worm dataset (Supplementary Table 2). GO analysis of these genes revealed enrichment in "ciliary and flagellar dependent cell motility", "movement", "axoneme assembly", and "microtubule bundle formation" (Supplementary Fig. 7b). These features are fundamental not only for locomotion and fluid movement in the ventral epidermis but also for sensory-neural functions given the presence of cilia in sensory neurons (Rompolas et al. 2013; Azimzadeh and Basquin 2016).

To explore whether loss of *soxB1-2* causes changes in gene expression that could be correlated with neoblast progeny adopting alternative differentiation fates, we also performed GO enrichment analysis on genes upregulated in *soxB1-2(RNAi)* animals. This upregulated gene set was enriched for metabolic processes, xenobiotic detoxification, potassium ion homeostasis, and endocytic pathways (Supplementary Fig. 7c). These categories are not strongly associated with known alternative lineage programs or fate switches. However, because these data were generated from bulk head tissue, the upregulated signatures could also reflect shifts in the relative abundance of specific cell types following loss of *soxB1-2*-expressing populations. Thus, we find no evidence from this analysis for a clear transcriptional shift toward alternative lineage programs. Consistent with a primary effect on neural gene expression programs, RNA-seq of head tissue revealed additional downregulated genes enriched for ciliated sensory neuron function that were not detected in whole-worm RNA-seq, suggesting a more specialized role for *soxB1-2* in these populations. We therefore next examined the genome-wide overlap between RNA-seq and ATAC-seq datasets to further define the regulatory effects of *soxB1-2* loss.

## Integrating ATAC-seq and RNA-seq identifies candidate downstream targets of *soxB1-2* in neural and ciliated neural cells

To uncover potential downstream targets of *soxB1-2*, we integrated our head region ATAC-seq and RNA-seq datasets. This combined approach allowed us to identify genes that not only exhibit changes in chromatin accessibility but also corresponding changes in transcriptional output. While we found consistent signals across both assays in some genes, others were detected only by one method, underscoring the complementary nature of chromatin and transcriptomic profiling.

While differences between ATAC-seq and RNA-seq datasets are expected due to the distinct regulatory layers they capture, their integration revealed a set of 31 genes exhibiting both decreased chromatin accessibility and transcriptional downregulation (Fig. 3a), highlighting them as strong candidates for direct downstream targets of *soxB1-2*. We assessed enrichment of the 31 candidate genes in neural and ciliated neural cell clusters and examined their co-expression with *soxB1-2* using single-cell RNA-seq datasets. Single-cell databases indicated most of the 31 genes were either enriched in the same neural and ciliated neural clusters where *soxB1-2* is also enriched or co-expressed with

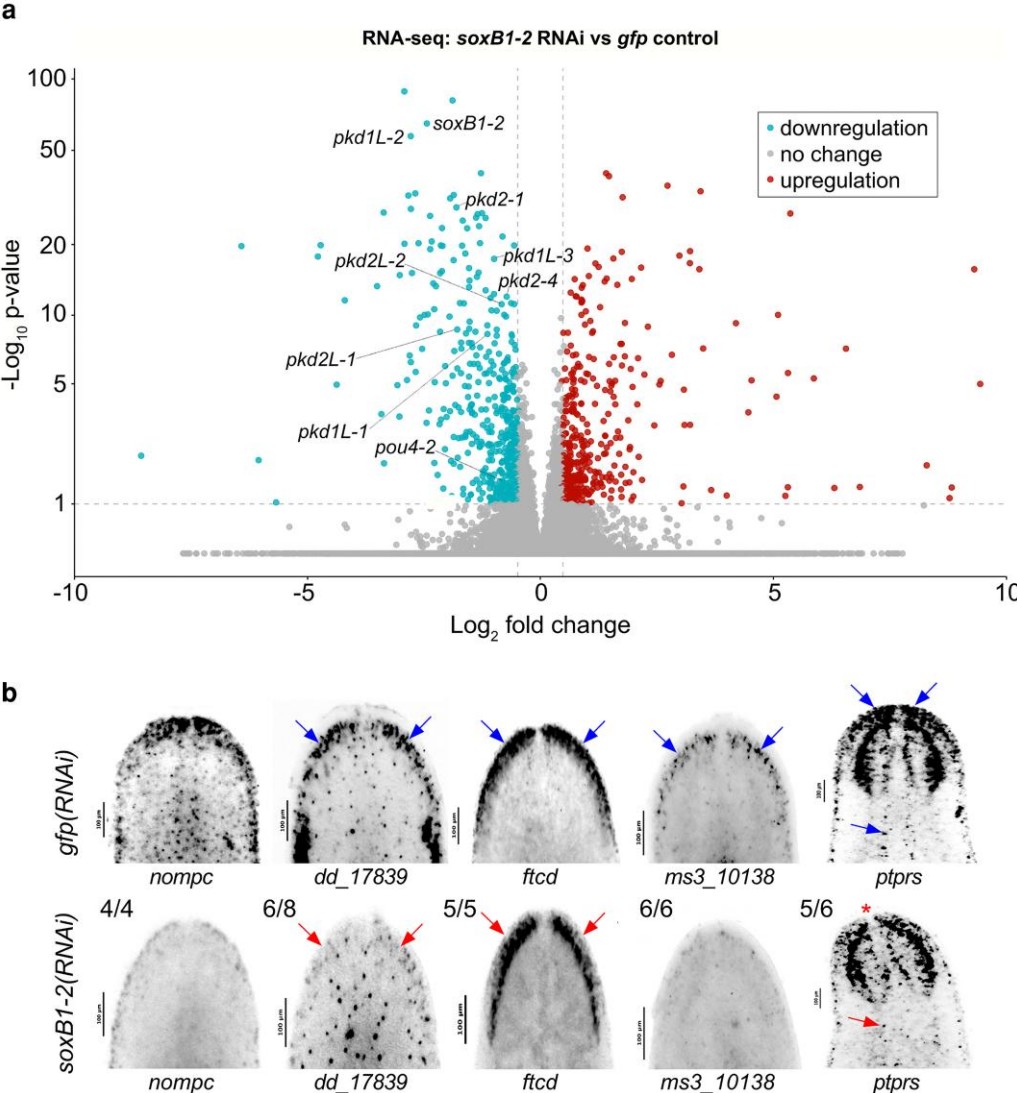

**Fig. 2.** Genes identified by ATAC-seq are regulated by *soxB1-2*. a) Volcano plot of differentially expressed genes in *gfp* control vs *soxB1-2* RNAi groups, with downregulated (blue) and upregulated (red) genes (FC ≥ 1.4, P-adj ≤ 0.1). Sample size: N = 4 and N = 6 for control and *soxB1-2* RNAi groups, respectively. b) In situ hybridization of putative sensory neural genes identified by RNA-seq following *soxB1-2* RNAi. Blue arrows indicate regions of in situ signal loss following *soxB1-2* RNAi. Red arrows mark anatomical regions with undetectable signal; red asterisk denotes loss of expression in the tip of the head. Sample size: For controls, N ≥ 3 worms were analyzed, and all samples showed consistent expression patterns. For RNAi-treated animals, the number of biological replicates (N) is indicated at the top left of each treatment group.

*soxB1-2* within these populations (Wurtzel et al. 2015; Fincher et al. 2018). Exceptions included *dd_380*, enriched in epidermal clusters, and *dd_8471* and *dd_8567*, which lacked specific cluster enrichment and exhibited broad co-expression across all cell types (Supplementary Table 3).

Among the genes identified, we were particularly interested in transcription factors, as these are likely to act as intermediate regulators within the broader *soxB1-2* gene regulatory network. By integrating ATAC-seq and RNA-seq, we could prioritize transcription factors that were either losing accessibility and downregulated at the chromatin and transcriptional levels or uniquely detected through chromatin accessibility changes, suggesting potential regulatory priming. This allowed us to expand the architecture of the *soxB1-2* pathway by highlighting candidate nodes acting downstream of *soxB1-2*.

Notably, transcription factors, including *castor*, were exclusively detected through ATAC-seq assays (Fig. 1d); they were not identified as differentially expressed in RNA-seq following *soxB1-2*

RNAi treatment (Table 1), potentially due to low transcript abundance or subtle expression changes. However, some transcription factors, such as *mecom* and *foxj1-5*, exhibited significant decreases in both chromatin accessibility and expression in *soxB1-2(RNAi)* animals (Fig. 3b). The *mecom* gene is a neural transcription factor that is expressed in the same neural and ciliated neural clusters as *soxB1-2* (Wurtzel et al. 2015; Fincher et al. 2018). ATAC-seq and RNA-seq peaks at the *mecom* locus in both *soxB1-2(RNAi)* and control animals are shown in Fig. 3b, demonstrating a significant reduction in chromatin accessibility following RNAi of *soxB1-2*.

GO analysis revealed distinct enrichment patterns between the ATAC-seq and the RNA-seq datasets (Fig. 3c, d). ATAC-seq enrichment was associated with direct neural functions, with "forebrain development" as the top term, alongside pathways such as "potassium ion transport", which is critical for neural processes like action potential generation and neurotransmitter release (Meir et al. 1999; Wang et al. 2023). Additionally, ATAC-seq revealed

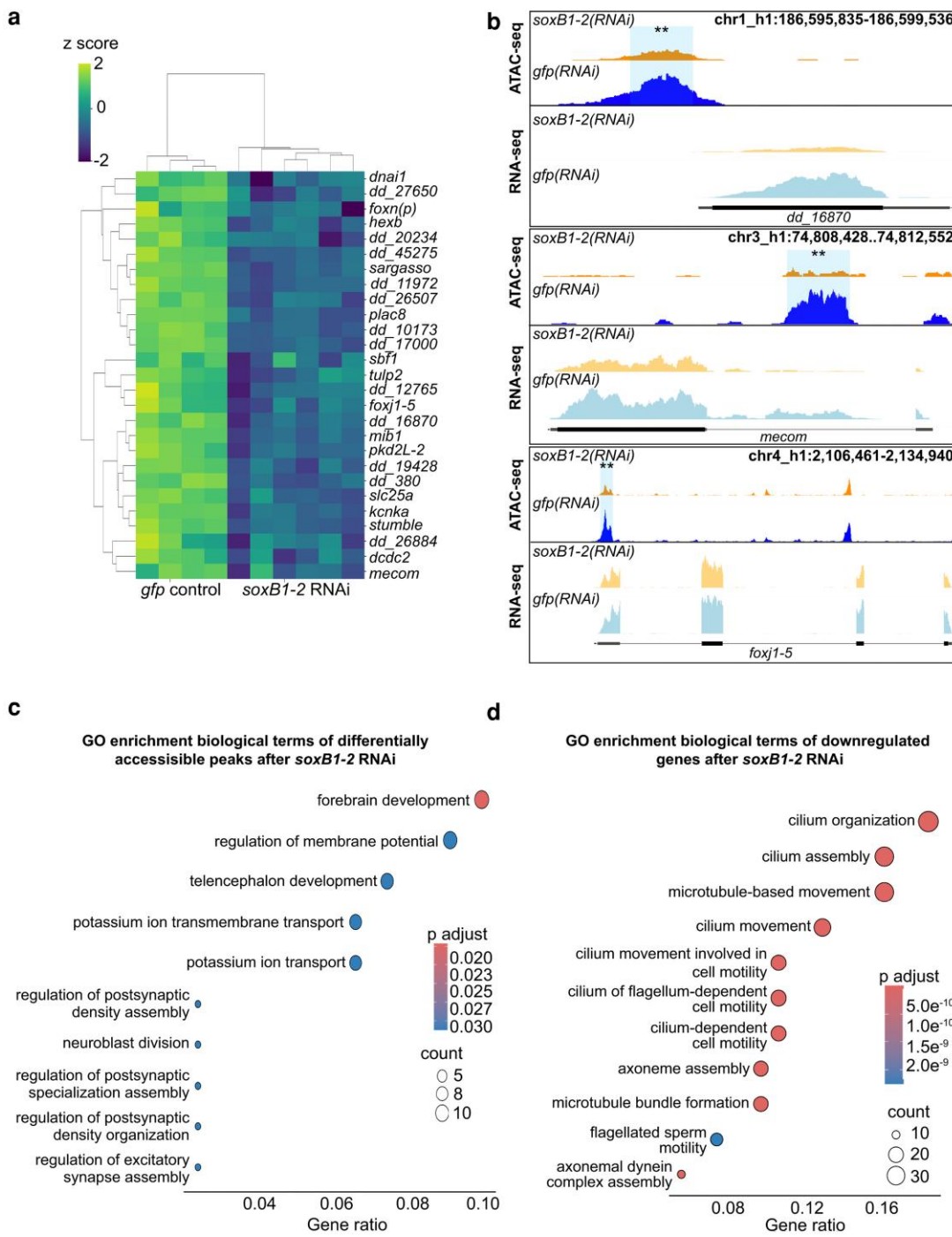

**Fig. 3.** ATAC-seq and RNA-seq reveal coordinated loss of chromatin accessibility and downregulation of neural genes. a) Heatmap of differentially expressed genes identified through RNA-seq (FC ≥ 1.4, P-adj ≤ 0.1) in *gfp* control vs *soxB1-2* RNAi groups. Z-scores indicate upregulation (2), no change (0), and downregulation (−2). Labeled genes are putative neural genes identified from both ATAC-seq and RNA-seq experiments. b) Genome tracks showing reduced chromatin accessibility and gene downregulation following *soxB1-2* RNAi at putative neural gene loci (*foxn(p)*, *foxj1*-5, and *stumble*) in S. *mediterranea*. c) GO term enrichment analysis of genes associated with reduced chromatin accessibility following *soxB1-2* knockdown. d) GO term enrichment analysis of downregulated genes in RNA-Seq experiments following *soxB1-2* knockdown.

enrichment for "synaptic development and assembly" (Fig. 3c). In contrast, RNA-seq data GO enrichment predominantly involved cilium-related processes, including movement, assembly, and development (Fig. 3d), consistent with our previous findings on gene expression associated with ciliated neurons and the epidermis (Ross et al. 2018). The enrichment of cilium-related terms aligns with the observed loss of chromatin accessibility and gene downregulation following *soxB1-2* RNAi in genes that have sensory or mechanosensory-related functions. Additionally, this gene enrichment was reflected in spatial expression data, as in situ hybridization of these potential *soxB1-2* target genes showed a decrease in expression in body regions strongly associated with ciliated sensory neurons (Figs. 1e, 2b, and 4 and Supplementary Table 3).

In addition to neural gene regulation, we observed overlapping chromatin accessibility loss and downregulation of multiple

genes enriched in epidermal progenitor and late epidermal lineages, as well as genes shared by both neural and epidermal populations. These include candidates potentially involved in epidermal fate determination, such as *dd_12765*, *foxj1-5*, *dd_13134*, and *dd_380* (Supplementary Table 3) (Vij et al. 2012; Fincher et al. 2018; Pascual-Carreras et al. 2021). The presence of both neural and epidermal signatures suggests that *soxB1-2* plays a broader role in ectodermal lineage identity. However, because this study primarily focuses on its function in neuronal lineages, our downstream analyses were concentrated on genes enriched in neural populations.

To validate that genes showing overlap in both datasets are regulated by *soxB1-2*, we cloned 14 of the 31 candidate targets and assessed their expression using in situ hybridization in both control and *soxB1-2* RNAi animals. All tested genes were expressed in the head margin and other regions enriched for sensory neurons, and all showed reduced or absent expression following *soxB1-2* knockdown (Fig. 4, Supplementary Fig. 8). These results support the model that *soxB1-2* activity regulates these genes and provide strong functional validation of our integrated RNA-seq and ATAC-seq approach for identifying *soxB1-2* targets in neural and ciliated neural populations. Furthermore, the validated genes include several with known or predicted roles in neural development or function (Fig. 3c). For example, *kcnka*, a potassium channel homolog, was previously shown to be

**Table 1.** Transcription factors with differential ATAC-seq accessibility but not differentially expressed in RNA-seq following *soxB1-2* RNAi.

| Gene | Transcription factor domain classification |
| --- | --- |
| *Pknox2 (dd_Smed_v6_8606)* | Homeobox |
| *Msh1 (dd_Smed_v6_18505)* | Homeobox |
| *Meox1 (dd_Smed_v6_12317)* | Homeobox |
| *Phtf2 (dd_Smed_v6_8465)* | Homeobox |
| *Castor (dd_Smed_v6_6778)* | Zinc finger |
| *Tcf-3 (dd_Smed_v6_8152)* | T-cell specific & High mobility group |
| *dd_Smed_v6_4717* | Ets-1 |
| *Tcf-2 (dd_Smed_v6_9140)* | T-cell specific & High mobility group |
| *Smad9 (dd_Smed_v6_6050)* | SMAD |
| *Islet1 (dd_Smed_v6_8820)* | LIM & Homeobox |

downregulated following *coe* RNAi, a neural transcription factor involved in nervous system development in *S. mediterranea* (Cowles et al. 2014).

## Distinct and overlapping roles of *mecom* and *castor* in mechanosensory signaling, ion transport, and ciliary regulation

Based on the significant loss of chromatin accessibility at the *mecom* and *castor* loci and their downregulation following *soxB1-2* RNAi, we investigated these genes as candidate downstream transcription factor targets of *soxB1-2* activity. While only *mecom* was significantly reduced in the *soxB1-2(RNAi)* RNA-seq dataset, we observed a reduction in expression for both genes in sensory regions by in situ hybridization (Figs. 1e and 4). Combined with their known roles in neural and sensory development in other model systems, these observations prompted us to define the regulatory targets of *mecom* and *castor* to better understand their contribution to the *soxB1-2*-dependent transcriptional network. To determine the optimal RNAi timepoints for transcriptomic analysis, we performed a time-course experiment across ten RNAi feedings and monitored *mecom* and *castor* expression levels. *mecom* transcript depletion was consistently observed by RNAi day 24 (Supplementary Fig. 9a, b). *castor* transcript loss occurred by day 37 (Supplementary Fig. 10a, b). RNA-seq analysis revealed that knockdown of *mecom* and *castor* resulted in the significant downregulation of 25 and 391 genes, respectively (Supplementary Tables 4 and 5). The relatively small number of *mecom*-dependent genes detected is consistent with its low baseline expression in a small number of cells (Fig. 4). Among the downregulated genes, *nompc* was the only mechanosensory receptor reduced in both *mecom* and *castor* RNAi samples (Figs. 5 and 6; Supplementary Figs. 9 and 10). *pkd2-1*, another known *soxB1-2* target involved in mechanosensation, was downregulated exclusively in the *mecom* dataset (Fig. 5a and Supplementary Fig. 9c). In contrast, *castor* knockdown led to downregulation of additional mechanosensory and ion channel-related genes, including *transient receptor potential ankyrin 1 (trpa1)* and *pkd* family members *pkd1L-1, pkd1L-3, and pkd2L-1* (Fig. 6a and Supplementary Fig. 10c), which were all previously implicated in *soxB1-2*-dependent mechanosensory regulation (Ross et al. 2024). In addition to mechanosensory and ion channel-related genes, *castor* RNAi resulted in the downregulation of numerous genes associated with ciliary

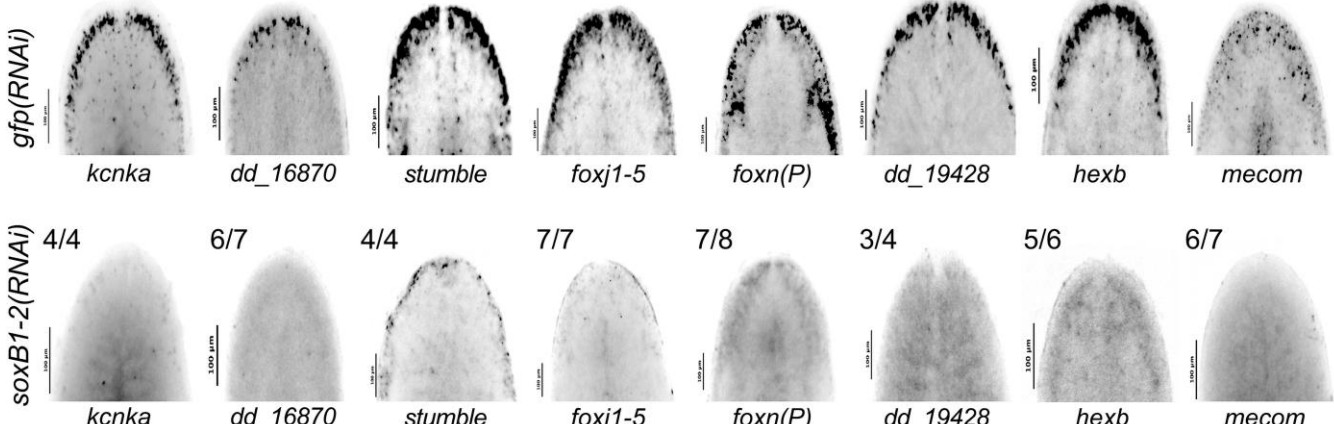

**Fig. 4.** In situ hybridization validation of neural genes identified in ATAC-seq and RNA-seq. a) In situ hybridization of neural genes expressed in the head rim, dorsal ciliated stripe, and periphery in *gfp* control (top) and *soxB1-2* RNAi (bottom) animals. Sample size: For controls, *N* ≥ 3 worms were analyzed, and all samples showed consistent expression patterns. For RNAi-treated animals, the number of biological replicates (*N*) is indicated at the top left of each treatment group.

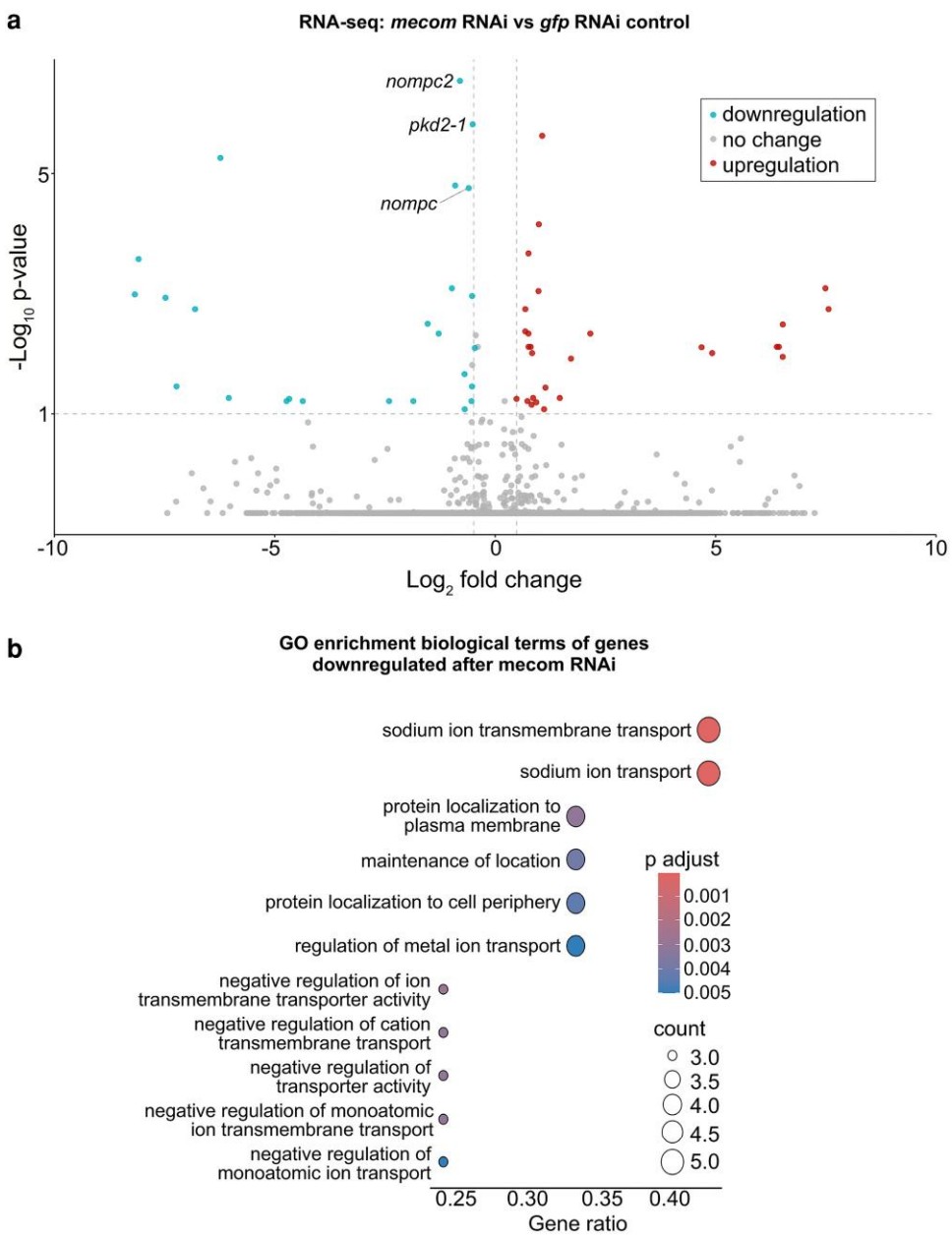

**Fig. 5.** *mecom* RNAi disrupts mechanosensory and ion channel-like gene expression. a) Volcano plot of differentially expressed genes in *gfp* control vs *mecom* RNAi groups, with downregulated (blue) and upregulated (red) genes (FC ≥ 1.4, P-adj ≤ 0.1). Sample size: N = 5 for control and *mecom* RNAi groups. b) GO term enrichment analysis of downregulated genes following *mecom* knockdown.

structure and function, including *dystonin* (*dst*) and multiple dynein component homologs. These findings suggest that while *mecom* and *castor* regulate distinct gene sets, both contribute to a mechanosensory and ion channel gene expression network downstream of *soxB1-2*.

GO enrichment analysis of downregulated genes in *mecom* RNAi revealed significant enrichment for "ion transporter activity", consistent with the functional roles of several affected genes. Some of these genes encode TRP channels, which detect mechanical stimuli and mediate ion flux across membranes, including *pkd2-1* and *nompc*. Additionally, genes involved in ion channel regulation, such as *cav3*, which encodes a homolog to human *caveolin 3*, were also downregulated (Supplementary Table 4), establishing a potential role for *mecom* in planarians in mechanosensory signaling and ion transport regulation. In contrast, although

multiple mechanosensory and ion channel-related genes were downregulated in the *castor* RNA-seq dataset following RNAi, enriched GO terms were primarily associated with ciliary functions, including "cilium movement", "microtubule bundle formation", "axoneme assembly", and "microtubule-based movement" (Fig. 6b). Given the role of cilia in planarian sensory neurons, this enrichment is consistent with *castor*'s potential role in sensory regulation. GO terms related to the detection of mechanical stimuli were also enriched, supporting the observed downregulation of mechanosensory genes, such as *nompc* and members of the PKD gene family, in planarians. Additionally, GO terms enriched among *castor*-regulated genes overlapped with those identified in the *soxB1-2* RNAi dataset (Figs. 3d and 6b). Together, these findings suggest that *castor* plays a key role in regulating genes involved in mechanosensation, ion transport, and ciliary function,

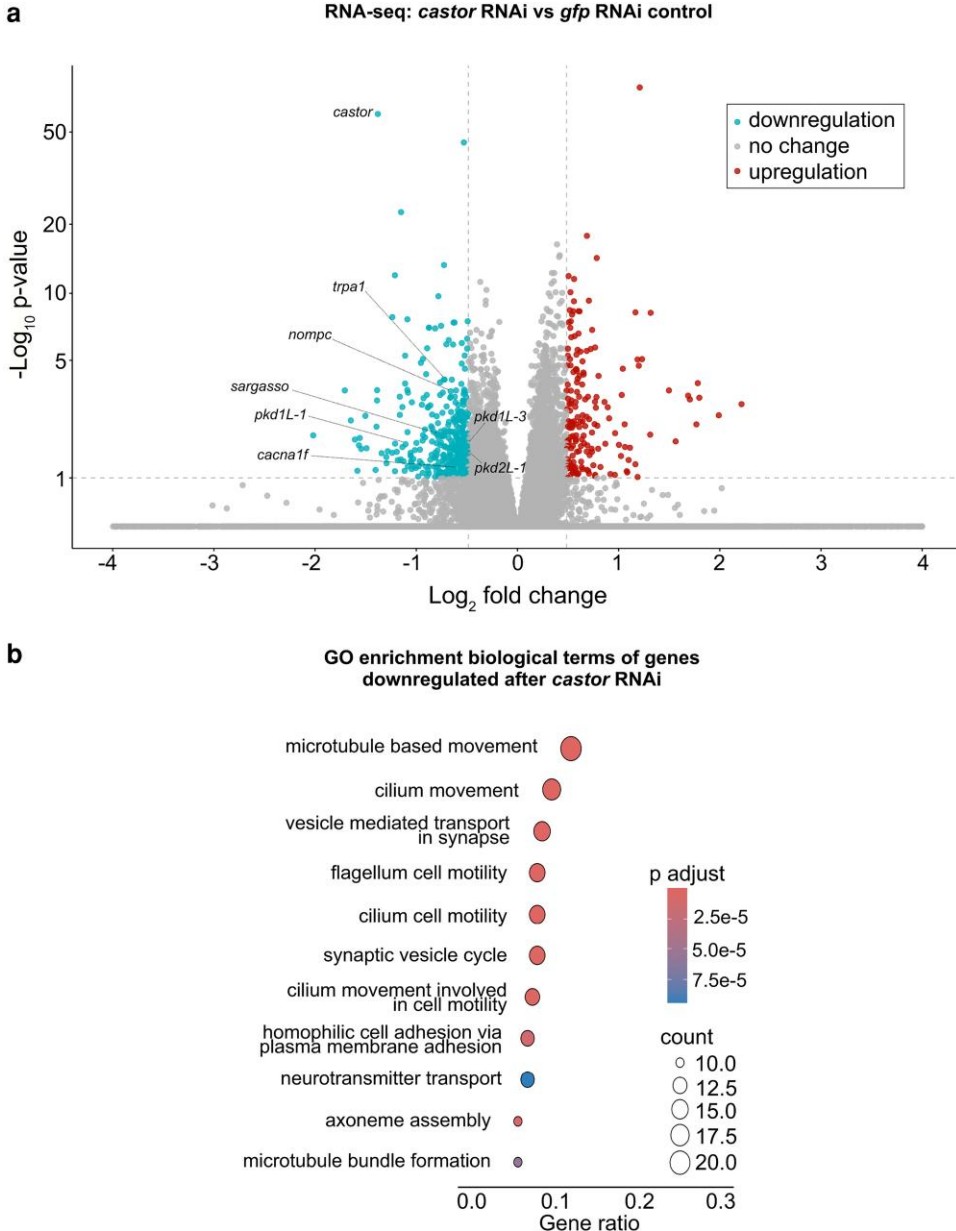

**Fig. 6.** *castor* RNAi disrupts genes involved in synapse and ciliary function. a) Volcano plot of differentially expressed genes in *gfp* control vs *castor* RNAi groups, with downregulated (blue) and upregulated (red) genes (FC ≥ 1.4, P-adj ≤ 0.1). Sample size: N = 5 for control and *castor* RNAi groups. b) GO term enrichment analysis of downregulated genes following *castor* knockdown.

positioning it as a critical component of the *soxB1-2* downstream regulatory network in planarian mechanosensory biology.

## Discussion

### Evidence for a transcriptional activator role of *soxB1-2* in chromatin remodeling during neural and epidermal specification

Planarians provide a powerful in vivo model for investigating adult neurogenesis and lineage specification. Leveraging this system, we have begun to dissect the role of *soxB1-2* in directing adult stem cell fate toward ectodermal lineages, with a particular focus on neural differentiation. Although prior studies established that *soxB1-2* promotes ectodermal identity in planarians (Ross et al. 2018), the molecular mechanisms underlying this process had not been explored. In both vertebrate and invertebrate systems,

SoxB1 family members are known to play critical roles in maintaining pluripotency, sustaining progenitor identity, and promoting ectodermal development (Buescher et al. 2002; Meulemans and Bronner-Fraser 2007; Kurtova et al. 2024). However, except for mammalian SOX2 homologs, few studies have addressed the mechanisms by which SoxB1 factors regulate these processes (Iwafuchi-Doi and Zaret 2014; Soufi et al. 2015). Here, we show that loss of *soxB1-2* leads to a marked reduction in chromatin accessibility at numerous neuronal and epidermal loci, some of which encode transcription factors. These findings suggest that *soxB1-2* may function as a transcriptional activator, initiating chromatin remodeling to prime progenitor cells for neural and epidermal differentiation.

Although our findings indicate that *soxB1-2* modulates chromatin accessibility and regulates downstream transcription factors including *mecom*, *foxj1-5*, and *castor*, we have not yet provided

direct biochemical evidence of SoxB1-2 binding to closed chromatin. However, motif enrichment analysis revealed that regions losing accessibility upon *soxB1-2* knockdown are significantly enriched for Sox-family motifs, particularly those resembling the SoxB1 subclass, consistent with a model in which SoxB1-2 directly establishes or maintains chromatin accessibility at its target loci. To definitively test this pioneer-like mechanism, future work will require generation of a SoxB1-2–specific antibody to enable chromatin profiling approaches such as CUT&RUN or ChIP–seq. Integrating these assays with ATAC-seq will allow us to distinguish direct vs indirect SoxB1-2 regulatory events and determine whether SoxB1-2 engages inaccessible chromatin in vivo.

Despite the expanding use of transcriptomic tools in planarians, genome-scale chromatin accessibility datasets remain scarce. Applying ATAC-seq in this system not only enables mechanistic investigation of soxB1-2 function but also broadens the epigenomic resources available for this emerging regenerative model. By generating high-quality chromatin accessibility profiles from control and *soxB1-2(RNAi)* animals, our study establishes a foundational dataset for investigating adult neural fate specification and offers a framework for integrating chromatin and transcriptional regulation in planarian stem cell biology.

## Mechanistic insights into *soxB1-2* function via ATAC-seq and RNA-seq integration post-RNAi

ATAC-seq has enhanced our understanding of the chromatin landscape in planarian neoblasts and throughout regeneration (Pascual-Carreras et al. 2023; Poulet et al. 2023; Ivankovic et al. 2024). When combined with RNAi-mediated perturbations, this approach has enabled detailed investigation of the epigenomic effects resulting from disruption of key regulators, including the stem cell marker *smedwi-2*, the chromatin remodeler BPTF (a core subunit of the Nucleosome Remodeling Factor, NuRF), and BAF complex components such as *brg1* and *smarcc2* (Li et al. 2021; Wiggans et al. 2023; Verma et al. 2025).

More broadly, the integration of ATAC-seq with RNA-seq has become a widely adopted strategy for linking chromatin accessibility to gene expression, with applications spanning neural development, cancer biology, and chromatin remodeling dynamics (Xu et al. 2023; Bai et al. 2024). Building on this, recent studies have begun applying this integrative strategy in planarians following RNAi (Bai et al. 2024). In our work, combining ATAC-seq and RNA-seq enabled a mechanistic analysis of *soxB1-2*-driven lineage specification and the regulation of its direct downstream targets. Although both assays were performed on head fragments, the number of biological replicates differed (ATAC-seq: $N = 2$ per condition; RNA-seq: $N = 4$ to 6 per condition), and we interpret cross-assay comparisons in light of these differences in statistical power. Nevertheless, the concordant loss of accessibility and gene expression at key neural regulators across assays supports a robust requirement for *soxB1-2* in neural lineage specification. We identified 31 genes that exhibited both differential chromatin accessibility and altered transcript levels following *soxB1-2* knockdown. These genes were either enriched in or co-expressed with *soxB1-2* across multiple cell types, including ciliated neural cells, general neural populations, early epidermal progenitors, and neoblasts (Supplementary Fig. 6 and Supplementary Table 3). Among these, ciliated neural cells showed the highest levels of enrichment and co-expression (Supplementary Table 3). These patterns support a model in which *soxB1-2* activity regulates ectodermal progenitor states through coordinated control of chromatin accessibility and gene expression (Fig. 7).

Furthermore, in situ hybridization of select candidate targets (Fig. 4, Supplementary Fig. 8) confirmed reduced or ablated expression of the putative target genes following *soxB1-2* RNAi, supporting their status as downstream, transcriptionally regulated genes. Interestingly, changes in chromatin accessibility following *soxB1-2* RNAi did not always correspond to changes in gene expression; some loci showed decreased accessibility without changes in transcript levels, while others exhibited altered expression despite stable accessibility. Independent validation of several of these loci (Figs. 1e and 2b) supports a model in which *soxB1-2* initiates chromatin remodeling that may precede or occur independently of transcriptional activation. This model aligns with findings from *Xenopus* spinal cord regeneration, where chromatin changes have been shown to prime cells for future gene expression (Kakebeen et al. 2020). These findings highlight the power of combining ATAC-seq and RNA-seq to reveal regulatory mechanisms that underlie lineage specification and transcriptional control during planarian tissue homeostasis or regeneration.

## Enhanced resolution of neural regulation via targeted analysis of head tissue

Similar to previous studies (Nakazawa et al. 2003; Roberts-Galbraith et al. 2016; Wang et al. 2016), using head tissue enriched for neural and ciliated neural cell populations improved our ability to detect expression changes in neurons. This spatially targeted sampling revealed gene expression changes in sensory neurons undetectable in previous whole-worm RNA-seq datasets (Ross et al. 2018), particularly within rare or spatially restricted cell types. For example, *mecom* exhibited significantly reduced chromatin accessibility and transcript levels following *soxB1-2* RNAi in the head tissue dataset. These changes were not observed in the whole-body RNA-seq dataset. In both our whole-body and head tissue RNA-seq datasets and head tissue ATAC-seq datasets, the samples represent heterogeneous cell populations, not isolated neural or *soxB1-2*$^+$ cells. This heterogeneity inherently introduces noise and likely obscures additional SoxB1-2 targets or changes in rare cell populations, even in our neural-enriched head tissue. Notably, we did not detect changes in chromatin accessibility or transcript levels in sensory populations marked by *pou4-2*, despite research in our lab showing strong downregulation following *soxB1-2* RNAi (McCubbin et al. 2025). This may reflect indirect regulation or a dilution effect due to the limited expression of *pou4-2* and the reduced sensitivity of bulk assays to capture changes in rare cell types. Future studies employing single-cell RNA-seq or FACS-based enrichment of *soxB1-2*$^+$ populations, pending the development of a validated antibody, will be critical for resolving these questions.

## *mecom* and *castor* are downstream of *soxB1-2* and are putative mechanosensory signaling, synapse, and ion transport regulators

Chromatin accessibility at regulatory regions near *mecom* and *castor* was markedly reduced following *soxB1-2* RNAi. Consistent with this, in situ hybridization showed a loss of *mecom* expression in cells and a substantial reduction in *castor* expression. These findings indicate that *mecom* and *castor* act downstream of *soxB1-2* and suggest that the corresponding neural cell populations are very likely not produced in the absence of *soxB1-2*, although further functional or cell-type-specific analyses will be needed to formally confirm their absence.

As cilia mediate mechanical sensing in planarians, these findings suggest that *mecom* and *castor* regulate mechanosensory populations downstream of *soxB1-2*, which is consistent with known roles of these genes in other organisms.

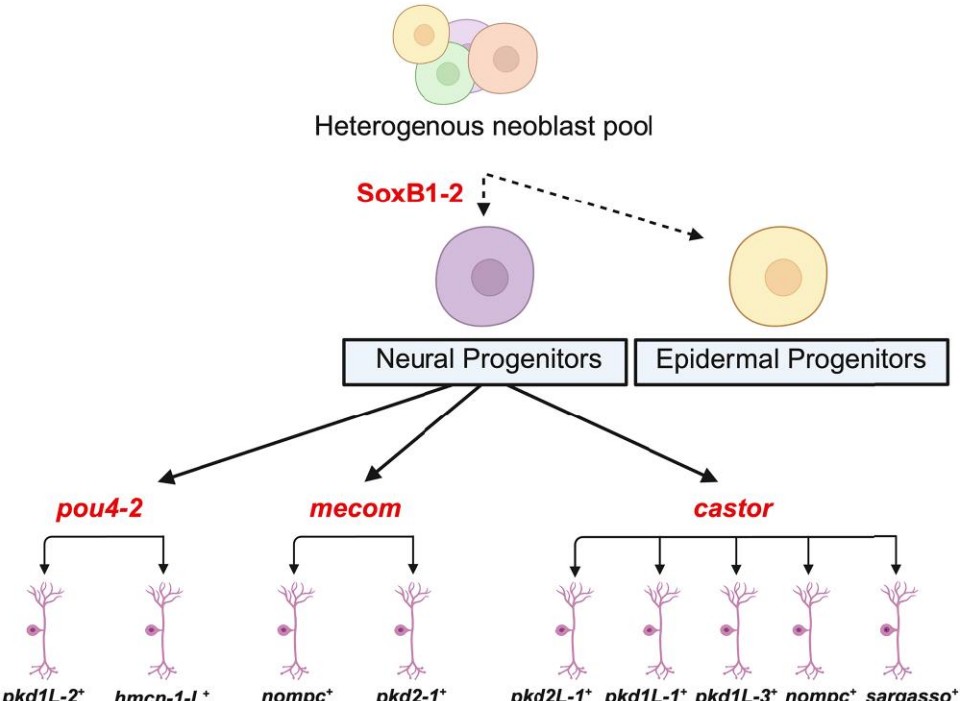

**Fig. 7.** Hypothetical model of SoxB1-2-regulated neuronal cell fates uncovered by integrating epigenomic and transcriptomic approaches. Previous work showed that *soxB1-2* is first expressed in a *piwi-1*[+] progenitor population that acquires epidermal or neuronal cell fates that terminally differentiate into ciliated sensory neurons or epidermis (Ross et al. 2018). Subsequent work has focused on understanding the mechanisms downstream of SoxB1-2 involved in specifying sensory neuron subclasses. McCubbin et al. (2025) found that the Pou4 homolog, *pou4-2*, regulates the differentiation of ciliated sensory neurons in the ciliated stripes. In this study, we show that *mecom* and *castor* also function in the *soxB1-2* regulatory network by regulating neuronal gene expression. Created in https://BioRender.com.

For example, Mecom promotes supporting cell proliferation and sensory hair cell differentiation in the mammalian cochlea, and Castor specifies early-born central complex neurons involved in sensory integration in Drosophila (Plath and Barron 2015; Chen et al. 2022; Dillon and Doe 2024). In the context of sensory neuron development, *mecom* overexpression has been shown to promote supporting cell proliferation and hair cell differentiation by modulating Tgf-β, Notch, and Wnt signaling pathways (Chen et al. 2022). The identification of *soxB1-2* downstream targets exhibiting both chromatin accessibility loss and transcriptional downregulation highlights its regulatory role in neural and ciliated neural populations, further supporting the involvement of genes like *mecom* in these developmental processes. Additional analyses will be required to delineate the precise cellular relationships among these factors.

Although transcript knockdown was confirmed by in situ hybridization prior to RNA-seq (Supplementary Figs. 9b and 10b), RNAi targeting *mecom* did not result in significant downregulation of *mecom* transcript levels in the *mecom* RNA-seq data. This likely reflects *mecom*'s low baseline expression or its spatially restricted enrichment at the head tip in both control and RNAi-treated animals. Because *mecom* is an uncharacterized gene in planarians, RNA-seq was performed on whole worms, which likely diluted transcripts originating from the head tip region. Future studies using spatially enriched sampling, such as head dissections, may improve detection sensitivity. Alternatively, the lack of statistical significance may reflect the use of stringent analytical thresholds.

In contrast, following *soxB1-2* RNAi, some downstream transcription factors exhibited clear reductions in both chromatin accessibility and transcript levels, whereas others showed loss of chromatin accessibility without meeting RNA-seq significance cutoffs. For example, *castor* did not meet RNA-seq significance

cutoffs in the *soxB1-2* RNA-seq dataset, despite clear accessibility changes at its locus. Such discrepancies between ATAC-seq and RNA-seq are not uncommon and may reflect regulatory complexity, temporal separation between chromatin remodeling and transcriptional output, or transcript stability that allows mRNA to persist despite reduced chromatin accessibility. Despite these caveats, the combined ATAC-seq and RNA-seq data support *mecom* and *castor* as components of a conserved regulatory program linking *soxB1-2* to sensory neuron differentiation (summarized in Fig. 7). Future work will be required to define the precise cellular contexts and regulatory mechanisms through which these factors function.

## Conclusion

Our study demonstrates that *soxB1-2* plays a critical role in lineage specification in S. *mediterranea* by modulating chromatin accessibility and gene expression. Integration of ATAC-seq and RNA-seq revealed that *soxB1-2* knockdown reduces chromatin accessibility at neural and epidermal loci, supporting the idea that *soxB1-2* functions as a pioneer-like transcription factor that primes chromatin for differentiation. We identified *mecom* and *castor* as downstream targets implicated in mechanosensory signaling, ion transport, and synapse formation, thereby expanding the regulatory network governed by *soxB1-2*. In addition to its mechanistic insights, this work establishes one of the few chromatin accessibility datasets in planarians, offering a foundational resource for the regeneration community and a platform for future hypothesis generation. Beyond planarian biology, these findings contribute to a broader understanding of adult neurogenesis and chromatin remodeling in regeneration-competent systems. The regulatory principles uncovered here provide a conceptual

framework that could inform stem cell reprogramming strategies in less regenerative organisms, including humans, where restoring neurogenic capacity remains a critical therapeutic challenge.

## Data availability

Strains and plasmids generated in this study are available from the authors upon reasonable request. Supplementary Table 1 includes all ATAC-seq differential accessibility results, associated statistics, motif enrichment analyses, and Pearson correlation matrices. Supplementary Tables 2, 4, and 5 contain complete differential gene expression results and statistics for *soxB1-2*, *mecom*, and *castor* RNAi experiments, respectively. Supplementary Table 6 lists all primer sequences and gene identifiers used for cloning. ATAC-seq data have been deposited in the NCBI Sequence Read Archive under BioProject accession PRJNA1292954. RNA-seq data have been deposited under BioProject accessions PRJNA1292856 and PRJNA1292967.
Supplemental material available at GENETICS online.

## Acknowledgments

This work is based in part on the doctoral dissertation of Mallory Cathell submitted to the University of California, San Diego, and San Diego State University in partial fulfillment of the requirements for the Doctor of Philosophy degree in Cell and Molecular Biology (Cathell 2025). We thank Arianna Farrar, Carina Real, and Jennifer Severance for their help with RNAi experiments, and Elizabeth Duncan and Prince Verma for helpful discussions and advice regarding ATAC-seq analysis.

## Funding

This study was supported by the Rees-Stealy Research Foundation and California Institute for Regenerative Medicine Grant EDUC4-12813 predoctoral and postdoctoral fellowships to M.C. and M.A.A., respectively, and NIH R01GM135657 to R.M.Z.

## Conflicts of interest

None declared.

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

*Editor: P. Newmark*