## [Peer Review File · Genetics]

SoxB1-Mediated Chromatin Remodeling Promotes Sensory Neuron Differentiation in Planarians

Mallory Cathell, Mohammad Auwal, Sarai Zepeda, Kelly Ross, and Ricardo Zayas

NOTE: The reviews and decision letters are unedited and appear as submitted by the reviewers.

In extremely rare instances and as determined by a Senior Editor or the EIC, portions of a review may be redacted. If a review is signed, the reviewer has agreed to no longer remain anonymous.

The review history appears in chronological order.

Review Timeline:

Submission Date:

2025-12-10

Accepted:

2025-12-18

EDITORIAL NOTE: This manuscript and reviews were transferred to GENETICS from Review Commons. The anonymized reviews and the authors' response to reviewer comments are included in this document.

Review
COMMONS

Revision 0

Review #1

1. Evidence, reproducibility and clarity:

Evidence, reproducibility and clarity (Required)

****Summary****

The authors of this interesting study take the approach of combing RNAi, RNA-seq and ATAC-seq to try to build a regulatory network surrounding the function of a planarian SoxB1 ortholog, broadly required for neural specification during planarian regeneration. They find a number of chromatin regions that differentially accessible (measured by ATAC-seq), associate these with potential genes by promity to the TSS. They then compare this set of genes with those that are differentially regulated (using RNA-seq), after SoxB1 RNAi mediated knockdown. This allows them the authors some focus on potential directly regulated targets of the planarian SoxB1. Two of these downstream targets, the mecom and castor transcription factors are then studied in greater detail.

****Major Comments.****

I have no suggestions for new experiments that fit sensibly with the scope of the current work. There are other analyses that could be appropriate with the ATAC-seq data, but may not make sense in the content of SoxB1 acting as pioneer factor.

I would like to see motif enrichment analysis under the set of peaks to see if SoxB1 is opening chromatin for a restricted set of other transcription factors to then bind. Much of this could be taken from Neuro et al, eLife 2022 (which also used ATAC-seq) and matched planarians TF families to likely binding motifs. This could add some breadth to the regulatory network. It could be revealing for example if downstream TF also help regulate other targets that SoxB1 makes available, this is pattern often seen for cell specification (as I am sure the authors are aware). Alternatively, it may reveal other candidate regulators.

Overall peak calling consistency with ATAC-sample would be useful to report as well, to give readers an idea of noise in the data. What was the correlation between samples?

While it is logical to focus on downregulated genes, it would also be interesting to look at upregulated genes in some detail. In simple terms would we expect to see the representation of an alternate set of fate decisions being made by neoblast progeny?

Can the authors be explicit about whether they have evidence for co-expression of SoxB1/castor and SoxB1/mecom? I could find this clearly and it would be important to be clear whether this basic piece of evidence is in place or not at this stage.

****Summary****

The authors of this interesting study take the approach of combing RNAi, RNA-seq and ATAC-seq to try to build a regulatory network surrounding the function of a planarian SoxB1 ortholog, broadly required for neural specification during planarian regeneration. They find a number of chromatin regions that differentially accessible (measured by ATAC-seq), associate these with potential genes by promity to the TSS. They then compare this set of genes with those that are differentially regulated (using RNA-seq), after SoxB1 RNAi

mediated knockdown. This allows them the authors some focus on potential directly regulated targets of the planarian SoxB1. Two of these downstream targets, the mecom and castor transcription factors are then studied in greater detail.

****Major Comments.****

N suggestions for new experiments that fit sensibly with the scope of the current work. There are other analyses that could be appropriate with the ATAC-seq data but may not make sense in the content of SoxB1 acting as pioneer factor. Overall, the study is executed very well, methods are sound, data and analysis well-presented and narrated, and the results placed in context. The experiments are clearly reproducible and can be built on, all data is accessible to others.

Motif enrichment analysis under the set of peaks to see if SoxB1 is opening chromatin for a restricted set of other transcription factors to then bind. Much of this could be taken from Neuro et al, eLife 2022 (which also used ATAC-seq) and matched planarians TF families to likely binding motifs. This could add some breadth to the regulatory network. It could be revealing for example if downstream TF also help regulate other targets that SoxB1 makes available, this is pattern often seen for cell specification (as I am sure the authors are aware). Alternatively, it may reveal other candidate regulators.

Overall peak calling consistency with ATAC-sample would be useful to report as well, to give readers an idea of noise in the data. What was the correlation between samples?

While it is logical to focus on downregulated genes, it would also be interesting to look at upregulated genes in some detail. In simple terms would we expect to see the representation of an alternate set of fate decisions being made by neoblast progeny?

Can the authors be explicit about whether they have evidence for co-expression of SoxB1/castor and SoxB1/mecom? I could find this clearly and it would be important to be clear whether this basic piece of evidence is in place, or not, at this stage.

****Minor comments.****

Formally loss of castor and mecom expression does mean these cells are absent, strictly the cell absence needs an independent method. It might be useful to clarify this with the evidence of be clear that cells are "very probably" not produced.

2. Significance:

Significance (Required)

Strengths and limitations.

The precise exploitation of the planarian system to identify potential targets, and therefore regulatory mechanisms, mediated by SoxB1 is an interesting contribution to the field. We know almost nothing about the regulatory mechanisms that allow regeneration and how these might have evolved, and this work is well-executed step in that direction.

Advance

The paper makes a clear advance in our understanding of an important process in animals (neural specification) and how this happens in the context in the context during an example of animal regeneration. The methods are state-of-the-art with respect to what is possible in the planarian system.

Audience

This will be of wide interest to developmental biologists, particularly those studying regeneration in planarians and other regenerative systems, and those who study comparative neurodevelopment.

Expertise

I have expertise in functional genomics in the context of stem cells and regeneration, particularly in the planarian model system

3. How much time do you estimate the authors will need to complete the suggested revisions:

Estimated time to Complete Revisions (Required)

(Decision Recommendation)

Between 1 and 3 months

Review #2

1. Evidence, reproducibility and clarity:

Evidence, reproducibility and clarity (Required)

Review - Cathell, et al (RC-2025-03195)

****Summary and Significance:****

Understanding regenerative neurogenesis has been difficult due to the limited amount of neurogenesis that occurs after injury in most animal species. Planarians, with their adult neurogenesis and robust post-injury response, allow us to get a glimpse into regenerative neurogenesis. The Zayas laboratory previously revealed a key role for SoxB1-2 in maintenance and regeneration of a broad set of sensory and peripheral neurons in the planarian body. SoxB1-2 also has a role in many epidermal fates. Their previous work left open the tempting possibility that SoxB1-2 acts as a very upstream regulator of epidermal and neuronal fates, potentially acting as a pioneer transcription factor within these lineages. In the manuscript currently under review, Cathell and colleagues use ATAC-Seq and RNA-Seq to investigate chromatin changes after SoxB1-2(RNAi). With the

experimental limitations in planarians, this is a strong first step toward testing their hypothesis that SoxB1-2 acts as a pioneer within a set of planarian lineages. Beyond these cell types, this work is also important because planarian cell fates often rely on a suite of transcription factors, but the nature of transcription factor cooperation has been much less well understood. Indeed, the authors do show that loss of SoxB1-2 by RNAi causes changes in a number of accessible regions of the genome; many of these chromatin changes correspond to changes in gene expression of genes nearby these peaks. The authors also examine in more detail two genes that have genomic and transcriptomic changes after SoxB1-2(RNAi), *mecom* and *castor*. The authors completed RNA-Seq on *mecom*(RNAi) and *castor*(RNAi) animals, identifying genes downregulated after loss of either factor that are also seen in SoxB1-2(RNAi). The results in this paper are rigorous and very well presented. I will share two major limitations of the study and some suggestions for addressing them, but this work may also be acceptable without those changes at some journals.

Limitation 1:

The paper aims to test the hypothesis that SoxB1-2 is a pioneer transcription factor. Observation that SoxB1-2(RNAi) leads to loss of many accessible regions in the chromatin supports the hypothesis. However, an alternate possibility is that SoxB1-2 leads to transcription of another factor that is a pioneer factor or a chromatin remodeling enzyme; in either of these cases, the accessibility peak changes may not be due to SoxB1-2 directly but due to another protein that SoxB1-2 promotes.

The authors describe how they can address this limitation in the future; in the meantime, is it known what the likely binding for SoxB1-2 would be (experimentally or based on homology)? If so, could the authors examine the relative abundance of SoxB1-2 binding sites in peaks that change after SoxB1-2(RNAi)? This could be compared to the abundance of the same binding sequence in non-changing peaks. Enrichment of SoxB1-2 binding sites in ATAC peaks that change after its RNAi would support the argument that chromatin changes are directly due to SoxB1-2.

Limitation 2:

The characterization of *mecom* and *castor* is somewhat preliminary relative to the deep work in the rest of the paper. I think this could be addressed with a few experiments. The authors could validate RNA-seq findings with ISH to show that cells are lost after reduction of either TF (this would support the model figure). The authors could also try to define whether loss of either TF causes behavioral phenotypes that might be similar to SoxB1-2(RNAi); this would be a second line of evidence that the TFs are downstream of key events in the SoxB1-2 pathway.

Other questions or comments for the authors:

Is it known how other Sox factors work as pioneer TFs? Are key binding partners known? I wondered if it would be possible to show that SoxB1-2 is co-expressed with the genes that encode these partners and/or if RNAi of these factors would phenocopy SoxB1-2. This is likely beyond the scope of this paper, but if the authors wanted to further support their argument about SoxB1-2 acting as a pioneer in planarians, this might be an additional way to do it.

This paper is one of few to use ATAC-Seq in planarians. First, I think the authors should make a bigger deal of their generation of a dataset with this tool! Second, it would be great to know whether the ATAC-Seq data (controls and/or RNAi) will be browsable in any planarian databases or in a new website for other scientists. I believe that in addition to the data being used to test hypotheses about planarians, the data could also be a huge hypothesis generating resource in the planarian community, so I would encourage the authors to both self-promote their contribution and make plans to share it as widely and useably as possible.

2. Significance:

Significance (Required)

This paper's strengths are that it addresses an important problem in regenerative biology in a rigorous manner. The writing and presentation of the data are excellent. The paper also provides excellent datasets that will be very useful to other researchers in the field. Finally, the work is one of, if not the first to examine how the action of one transcription factor in planarians leads to changes in the cellular and chromatin environment that could then be acted upon by subsequent factors. This is an important contribution to the planarian field, but also one that will be useful for other developmental neuroscientists and regenerative biologists.

I described a couple of limitations in the review above, but the strengths outweigh the weaknesses.

3. How much time do you estimate the authors will need to complete the suggested revisions:

Estimated time to Complete Revisions (Required)

(Decision Recommendation)

Between 1 and 3 months

Review #3

1. Evidence, reproducibility and clarity:

Evidence, reproducibility and clarity (Required)

The authors investigated the role of soxB1-2 in planarian neural and epidermal lineage specification. Using ATAC-seq and RNA-seq from head fragments after soxB1-2 RNAi, they identified regions of decreased chromatin accessibility and reduced gene expression, demonstrating that soxB1-2 induces neural and sensory programs. Integration of the datasets yielded 31 overlapping candidate targets correlating ATAC-seq and RNA-seq. Downstream analyses of transcription factors that had either/or differentially accessible regulatory region or showed differential expression (castor and mecom) implicated these transcription factors in mechanosensory and ciliary modules. The authors combined additional techniques, such as in situ hybridization to support the observations based on the ATACseq/RNAseq data. The manuscript is clearly written as well as data presentation in the main and supplementary figures. The major claim of the manuscript is that SoxB1-2 is likely a pioneer

transcription factor that alters the accessibility of the chromatin, which if true, would be one of the first demonstrations of direct transcriptional regulation in planarians. As described below, I am not certain that this interpretation of the data is more valid than alternative interpretations.

****Major comments****

1. Direct vs. indirect regulation. The current analysis does not distinguish between direct and indirect soxB1-2 targets, therefore, this analysis cannot indicate whether soxB1-2 functions as a pioneer transcription. ATAC-seq and RNA-seq, as performed here, do not determine whether reduced accessibility or downregulation of gene expression represents a change within existing cells or a reduction in the proportion of specific cell types in the libraries produced. This limitation should be explicitly recognized where causal statements are made. In fact, several pieces of information strongly suggest that indirect effects are abundant in the data: (1) the observed loss of accessibility and gene expression in late epidermal progenitors likely represent indirect effects, indicating that within the timeframe of the experiment, it is impossible (using these techniques) to distinguish between the scenarios. (2) The finding that castor knockdown reduces soxB1-2 expression likely reflects population loss rather than direct regulation, given overlapping expression domains. This further illustrates the difficulty in inferring directionality from such datasets.

In order to provide evidence for a more direct association between soxB1-2 and the differentially accessible chromatin regions, a sequence (e.g., motif) analysis would be required. Other approaches to infer direct regulation would have been useful, but they are not available in planarians to the best of my knowledge.

2. Evidence for pioneer activity. The authors correctly acknowledge that they do not present direct evidence of soxB1-2 binding or chromatin opening. However, the section title in the Discussion could be interpreted as implying otherwise. The claim of pioneer activity should remain explicitly tentative until supported (at least) by motif or binding data.

3. Replication and dataset comparability. Both ATAC-seq and soxB1-2 RNA-seq were performed on head fragments, but the number of replicates differ between assays (ATAC-seq n=2 per group, RNA-seq n=4-6). This is of course acceptable, but when interpreting the results, it should be taken into consideration that the statistical power is different when using data collected using different techniques and having a varied number of replicates.

****Minor comments****

"Thousands of accessible chromatin sites". Please state the number of peaks and the thresholds for calling them. Ensure consistency between text (264 DA peaks) and Figure 1 legend (269 DA peaks).

Specify the y-axis normalization units in all coverage plots.

Clarify replicate numbers consistently in the text and figure legends.

Referees cross commenting

The reviews are highly consistent. They recognize the value of the work, and raise similar points. The main shared view is that the current data do not distinguish direct from indirect effects, and claims about pioneer activity should be softened, and further analysis of the differentially accessible peaks could strengthen the link between SoxB1-2 and the chromatin changes.

- I don't think that it's necessary to further characterize experimentally mecom or castor (as suggested), but of course that it could have value.

2. Significance:

Significance (Required)

General assessment. The study offers valuable observations by combining chromatin and transcriptional analysis of planarian neural differentiation. The integration with in situ validation convincingly demonstrates effects on neural tissues and provides a solid resource for future functional work. However, mechanistic

interpretation remains limited, partly because of technical limitations of the system. The data support an important role for soxB1-2 in neural and epidermal lineage regulation, but not direct binding or chromatin-opening activity. The authors have previously published analysis of soxB1-2 in planarians, so the addition of ATAC-seq data contributes to solving another piece of the puzzle.

Advance. This is one of the first studies to couple ATAC-seq and RNA-seq in planarian tissue to dissect regulatory logic during regeneration. It identifies new candidate regulators of sensory and epidermal differentiation and identifies soxB1-2 as a likely upstream factor in ectodermal lineage networks. The work extends previous studies on soxB1-2 activity and neural cell production by integrating chromatin and transcriptional layers. In that respect the results are very solid, although the study remains correlative at the mechanistic level.

Audience. This work will potentially interest researchers interested in regeneration and transcriptional networks. The datasets and gene lists will be valuable references for follow-up studies on planarian ectodermal lineages, and therefore will appeal to this community.

3. How much time do you estimate the authors will need to complete the suggested revisions:

Estimated time to Complete Revisions (Required)

(Decision Recommendation)

Between 1 and 3 months

Manuscript number: RC-2025-03195R

Point-by-Point Response to Reviewers

We thank the reviewers for their thoughtful and constructive evaluations, which have helped us substantially improve the clarity, rigor, and balance of our manuscript. We are grateful for their recognition that our integrated ATAC-seq and RNA-seq analyses provide a valuable and technically sound contribution to understanding *soxB1-2* function and regenerative neurogenesis in planarians.

We have carefully addressed the reviewers' major points as follows:

- 1. Direct versus indirect regulation by SoxB1-2:** In the revision, we explicitly acknowledge the limitations of inferring direct regulation from our current datasets and have revised statements throughout the Results and Discussion to emphasize that our findings are correlative.
- 2. Evidence for pioneer activity:** Although the pioneer role of SoxB1 transcription factors is well established in other systems, we agree that additional binding or motif data would be required to formally demonstrate SoxB1-2 pioneer function. Accordingly, we performed motif analysis and revised the text throughout to frame SoxB1-2's proposed role as *consistent with*, rather than *demonstrating* transcriptional activator activity.
- 3. Motif enrichment and downstream regulatory interactions:** In response to Reviewer #1's suggestion, we have included a new motif enrichment analysis in the supplement to contextualize possible co-regulators within the SoxB1-2 network.
- 4. Data reproducibility and peak-calling consistency:** We have included sample correlations and peak overlaps for ATAC-seq samples in the revision, providing a clearer assessment of reproducibility.
- 5. Clarification of co-expression and downstream targets:** We included co-expression plots for *soxB1-2* with *mecom* and *castor* in the supplemental materials. These plots were generated from previously published scRNA-seq data and demonstrate that cells expressing *soxB1-2* also express *mecom* and *castor*.

We appreciate the reviewers' recognition that our methods are rigorous and our data accessible. We have incorporated all major revisions suggested and believe have strengthened the manuscript's precision, interpretations, and conclusions. Below, we respond to each comment in detail.

Reviewer #1 (Evidence, reproducibility and clarity (Required)):

Summary

The authors of this interesting study take the approach of combining RNAi, RNA-seq and ATAC-seq to try to build a regulatory network surrounding the function of a planarian SoxB1 ortholog, broadly required for neural specification during planarian regeneration. They find a

number of chromatin regions that differentially accessible (measured by ATAC-seq), associate these with potential genes by proximity to the TSS. They then compare this set of genes with those that are differentially regulated (using RNA-seq), after SoxB1 RNAi mediated knockdown. This allows them the authors some focus on potential directly regulated targets of the planarian SoxB1. Two of these downstream targets, the *mecom* and *castor* transcription factors are then studied in greater detail.

Major Comments

I have no suggestions for new experiments that fit sensibly with the scope of the current work. There are other analyses that could be appropriate with the ATAC-seq data, but may not make sense in the content of SoxB1 acting as pioneer factor.

I would like to see motif enrichment analysis under the set of peaks to see if SoxB1 is opening chromatin for a restricted set of other transcription factors to then bind. Much of this could be taken from Neuro et al, eLife 2022 (which also used ATAC-seq) and matched planarians TF families to likely binding motifs. This could add some breadth to the regulatory network. It could be revealing for example if downstream TF also help regulate other targets that SoxB1 makes available, this is pattern often seen for cell specification (as I am sure the authors are aware). Alternatively, it may reveal other candidate regulators.

Thank you for this suggestion. We agree with the reviewers that this analysis should be done. We ran the motif enrichment analysis using the same methods as outlined in Neuro et al. eLife, 2022. We have included a new motif enrichment analysis in the supplement to contextualize possible co-regulators within the SoxB1-2 network.

Overall peak calling consistency with ATAC-sample would be useful to report as well, to give readers an idea of noise in the data. What was the correlation between samples?

Excellent point. In response to this comment, we ran a Pearson correlation test on replicates within *gfp* and *soxB1-2* RNAi replicates to get an idea of overall correlation between replicates. Additionally, we calculated percent overlap of peaks for biological replicates and between treatment groups.

While it is logical to focus on downregulated genes, it would also be interesting to look at upregulated genes in some detail. In simple terms would we expect to see the representation of an alternate set of fate decisions being made by neoblast progeny?

This is also an important point that we considered but initially did not pursue it due to the lack of tools to test upregulated gene function. However, the reviewer is correct that this is straightforward to perform computationally. Thus, we have performed Gene Ontology analysis on the upregulated genes in all RNA-seq datasets (*soxB1-2* RNAi, *mecom* RNAi, and *castor* RNAi). Both *mecom* and *castor* datasets did not reveal enrichment within the upregulated portion of the dataset. Genes upregulated after *soxB1-2* RNAi were enriched for metabolic, xenobiotic detoxification, potassium homeostasis, and endocytic programs. Rather than indicating a shift toward alternative lineages, including non-ectodermal fates, these signatures are consistent with stress-responsive and homeostatic programs activated

following loss of *soxB1-2*. We did not detect enrichment patterns strongly associated with alternative cell fates. We conclude that this analysis does not formally exclude potential shifts in lineage-specific transcriptional programs, but does support our hypothesis that *soxB1-2* functions as a transcriptional activator.

Can the authors be explicit about whether they have evidence for co-expression of SoxB1/*castor* and SoxB1/*mecom*? I could find this clearly and it would be important to be clear whether this basic piece of evidence is in place or not at this stage.

We included co-expression plots for *soxB1-2* with *mecom* and *castor* in the supplemental material. These plots were generated from previously published scRNA-seq data and demonstrate that cells expressing *soxB1-2* also express *mecom* and *castor*. We have not done experiments showing co-expression via in situ at this time.

Minor comments

Formally loss of *castor* and *mecom* expression does mean these cells are absent, strictly the cell absence needs an independent method. It might be useful to clarify this with the evidence of be clear that cells are "very probably" not produced.

We agree that loss of *castor* and *mecom* expression does not formally demonstrate the physical absence of these cells, and that independent methods would be required to definitively confirm their loss. In response, we have revised our wording to indicate that *castor*- and *mecom*-expressing cells are very likely not being produced, rather than stating that they are absent.

Reviewer #1 (Significance (Required)):

Significance

Strengths and limitations.

The precise exploitation of the planarian system to identify potential targets, and therefore regulatory mechanisms, mediated by SoxB1 is an interesting contribution to the field. We know almost nothing about the regulatory mechanisms that allow regeneration and how these might have evolved, and this work is well-executed step in that direction.

Advance

The paper makes a clear advance in our understanding of an important process in animals (neural specification) and how this happens in the context in the context during an example of animal regeneration. The methods are state-of-the-art with respect to what is possible in the planarian system.

Audience

This will be of wide interest to developmental biologists, particularly those studying regeneration in planarians and other regenerative systems, and those who study comparative neurodevelopment.

Expertise

I have expertise in functional genomics in the context of stem cells and regeneration, particularly in the planarian model system

Reviewer #2 (Evidence, reproducibility and clarity (Required)): Review - Cathell, et al (RC-2025-03195)

Summary and Significance:

Understanding regenerative neurogenesis has been difficult due to the limited amount of neurogenesis that occurs after injury in most animal species. Planarians, with their adult neurogenesis and robust post-injury response, allow us to get a glimpse into regenerative neurogenesis. The Zayas laboratory previously revealed a key role for SoxB1-2 in maintenance and regeneration of a broad set of sensory and peripheral neurons in the planarian body. SoxB1-2 also has a role in many epidermal fates. Their previous work left open the tempting possibility that SoxB1-2 acts as a very upstream regulator of epidermal and neuronal fates, potentially acting as a pioneer transcription factor within these lineages. In the manuscript currently under review, Cathell and colleagues use ATAC-Seq and RNA-Seq to investigate chromatin changes after SoxB1-2(RNAi). With the experimental limitations in planarians, this is a strong first step toward testing their hypothesis that SoxB1-2 acts as a pioneer within a set of planarian lineages. Beyond these cell types, this work is also important because planarian cell fates often rely on a suite of transcription factors, but the nature of transcription factor cooperation has been much less well understood. Indeed, the authors do show that loss of SoxB1-2 by RNAi causes changes in a number of accessible regions of the genome; many of these chromatin changes correspond to changes in gene expression of genes nearby these peaks. The authors also examine in more detail two genes that have genomic and transcriptomic changes after SoxB1-2(RNAi), *mecom* and *castor*. The authors completed RNA-Seq on *mecom*(RNAi) and *castor*(RNAi) animals, identifying genes downregulated after loss of either factor that are also seen in SoxB1-2(RNAi). The results in this paper are rigorous and very well presented. I will share two major limitations of the study and some suggestions for addressing them, but this work may also be acceptable without those changes at some journals.

Limitation 1:

The paper aims to test the hypothesis that SoxB1-2 is a pioneer transcription factor. Observation that SoxB1-2(RNAi) leads to loss of many accessible regions in the chromatin supports the hypothesis. However, an alternate possibility is that SoxB1-2 leads to transcription of another factor that is a pioneer factor or a chromatin remodeling enzyme; in either of these cases, the accessibility peak changes may not be due to SoxB1-2 directly but due to another protein that SoxB1-2 promotes. The authors describe how they can address this limitation in the future; in the meantime, is it known what the likely binding for SoxB1-2 would be (experimentally or based on homology)? If so, could the authors examine the relative abundance of SoxB1-2 binding sites in peaks that change after SoxB1-2(RNAi)? This could be compared to the abundance of the same binding sequence in non-changing peaks. Enrichment of SoxB1-2 binding sites in ATAC peaks that change after its RNAi would support the argument that chromatin changes are directly due to SoxB1-2.

We appreciate the feedback and agree that distinguishing between direct SoxB1-2 pioneer activity and indirect effects mediated through downstream regulators is an important consideration. While we did not perform a direct abundance analysis of potential chromatin-remodeling cofactors, we conducted a motif enrichment analysis following the approach of Neuro et al. (eLife, 2022), comparing control and *soxB1-2(RNAi)* peak sets. This analysis revealed that Sox-family motifs, particularly SoxB1-like motifs, were among the most enriched in regions that remain accessible in control animals relative to *soxB1-2(RNAi)* animals, consistent with a model in which SoxB1-2 directly contributes to establishing or maintaining accessibility at these loci. We have now included this analysis in the supplemental materials to further contextualize potential co-regulators and transcriptional partners within the SoxB1-2 regulatory network. We agree and acknowledge in the report that future studies assessing chromatin remodeling factor expression and abundance will be valuable to definitively separate direct and indirect pioneer activity.

Limitation 2:

The characterization of *mecom* and *castor* is somewhat preliminary relative to the deep work in the rest of the paper. I think this could be addressed with a few experiments. The authors could validate RNA-seq findings with ISH to show that cells are lost after reduction of either TF (this would support the model figure). The authors could also try to define whether loss of either TF causes behavioral phenotypes that might be similar to SoxB1-2(RNAi); this would be a second line of evidence that the TFs are downstream of key events in the SoxB1-2 pathway.

Thank you for this suggestion. We agree that additional validation of the *mecom* and *castor* RNA-seq results and further phenotypic characterization would strengthen this section. We are currently conducting in situ hybridization experiments to validate transcriptional changes in *mecom* and *castor* using the same experimental framework applied to *soxB1-2* downstream candidates. We anticipate completing these studies within the next three months and will incorporate the results into future work.

Regarding behavioral phenotypes, we performed preliminary screening for robust behavioral responses, including mechanosensory responses, but did not observe overt defects. However, the lack of established, standardized behavioral assays in planarians presents a current limitation; such assays need to be developed *de novo*, and predicting specific behavioral phenotypes in advance remains challenging. We fully agree that functional behavioral assays represent an important next step and are actively exploring strategies to systematically develop and implement them going forward.

Other questions or comments for the authors:

Is it known how other Sox factors work as pioneer TFs? Are key binding partners known? I wondered if it would be possible to show that SoxB1-2 is co-expressed with the genes that encode these partners and/or if RNAi of these factors would phenocopy SoxB1-2. This is likely

beyond the scope of this paper, but if the authors wanted to further support their argument about SoxB1-2 acting as a pioneer in planarians, this might be an additional way to do it.

In other systems, Sox pioneer factors often act together with POU family transcription factors (for example, Oct4 and Brn2) and PAX family members such as Pax6. In planarians, a POU homolog (*pou-p1*) is expressed in neoblasts and may represent an interesting candidate co-factor for future investigation in the context of SoxB1-2 pioneer activity. We have also previously examined the relationship between SoxB1-2 and the POU family transcription factors *pou4-1* and *pou4-2*. Although RNAi of these factors does not fully phenocopy *soxB1-2* knockdown, *pou4-2(RNAi)* results in loss of mechanosensation, suggesting that downstream POU factors may contribute to aspects of neural function regulated by SoxB1-2 (McCubbin et al. eLife 2025). We agree that co-expression and functional interaction studies with these candidates would be highly informative, and we view this as an exciting future direction beyond the scope of the current manuscript.

This paper is one of few to use ATAC-Seq in planarians. First, I think the authors should make a bigger deal of their generation of a dataset with this tool! Second, it would be great to know whether the ATAC-Seq data (controls and/or RNAi) will be browsable in any planarian databases or in a new website for other scientists. I believe that in addition to the data being used to test hypotheses about planarians, the data could also be a huge hypothesis generating resource in the planarian community, so I would encourage the authors to both self-promote their contribution and make plans to share it as widely and useably as possible.

Thank you very much for this encouraging feedback. We appreciate the suggestion and have strengthened the text to emphasize the significance of generating this ATAC-seq resource for the planarian field. We agree that these datasets represent a valuable community resource and are committed to making all control and *soxB1-2(RNAi)* ATAC-seq data publicly accessible.

Reviewer #2 (Significance (Required)):

This paper's strengths are that it addresses an important problem in regenerative biology in a rigorous manner. The writing and presentation of the data are excellent. The paper also provides excellent datasets that will be very useful to other researchers in the field. Finally, the work is one of, if not the first to examine how the action of one transcription factor in planarians leads to changes in the cellular and chromatin environment that could then be acted upon by subsequent factors. This is an important contribution to the planarian field, but also one that will be useful for other developmental neuroscientists and regenerative biologists.

I described a couple of limitations in the review above, but the strengths outweigh the weaknesses.

Reviewer #3 (Evidence, reproducibility and clarity (Required)):

The authors investigated the role of *soxB1-2* in planarian neural and epidermal lineage specification. Using ATAC-seq and RNA-seq from head fragments after *soxB1-2* RNAi, they identified regions of decreased chromatin accessibility and reduced gene expression, demonstrating that *soxB1-2* induces neural and sensory programs. Integration of the datasets yielded 31 overlapping candidate targets correlating ATAC-seq and RNA-seq. Downstream

analyses of transcription factors that had either/or differentially accessible regulatory region or showed differential expression (castor and mecom) implicated these transcription factors in mechanosensory and ciliary modules. The authors combined additional techniques, such as in situ hybridization to support the observations based on the ATACseq/RNAseq data. The manuscript is clearly written as well as data presentation in the main and supplementary figures. The major claim of the manuscript is that SoxB1-2 is likely a pioneer transcription factor that alters the accessibility of the chromatin, which if true, would be one of the first demonstrations of direct transcriptional regulation in planarians. As described below, I am not certain that this interpretation of the data is more valid than alternative interpretations.

Major comments

1. Direct vs. indirect regulation. The current analysis does not distinguish between direct and indirect soxB1-2 targets, therefore, this analysis cannot indicate whether soxB1-2 functions as a pioneer transcription. ATAC-seq and RNA-seq, as performed here, do not determine whether reduced accessibility or downregulation of gene expression represents a change within existing cells or a reduction in the proportion of specific cell types in the libraries produced. This limitation should be explicitly recognized where causal statements are made. In fact, several pieces of information strongly suggest that indirect effects are abundant in the data: (1) the observed loss of accessibility and gene expression in late epidermal progenitors likely represent indirect effects, indicating that within the timeframe of the experiment, it is impossible (using these techniques) to distinguish between the scenarios. (2) The finding that castor knockdown reduces soxB1-2 expression likely reflects population loss rather than direct regulation, given overlapping expression domains. This further illustrates the difficulty in inferring directionality from such datasets.

In order to provide evidence for a more direct association between soxB1-2 and the differentially accessible chromatin regions, a sequence(e.g., motif) analysis would be required. Other approaches to infer direct regulation would have been useful, but they are not available in planarians to the best of my knowledge.

We agree that distinguishing between direct SoxB1-2 pioneer activity and indirect chromatin changes mediated by downstream factors is an important consideration. As suggested, examining the enrichment of SoxB1-2 binding motifs in regions that lose accessibility following *soxB1-2(RNAi)* can provide supporting evidence for direct regulation.

While we did not conduct a direct abundance analysis of all potential chromatin-remodeling cofactors, we performed a motif enrichment analysis following the methodology of Neuro et al. (eLife, 2022), comparing control-specific and *soxB1-2(RNAi)*-specific accessible peak sets. Consistent with a direct role for SoxB1-2 in chromatin regulation, Sox-family motifs, particularly SoxB1-like motifs, were among the most significantly enriched in regions that maintain accessibility in control animals relative to *soxB1-2(RNAi)* animals.

2. Evidence for pioneer activity. The authors correctly acknowledge that they do not present direct evidence of soxB1-2 binding or chromatin opening. However, the section title in the

Discussion could be interpreted as implying otherwise. The claim of pioneer activity should remain explicitly tentative until supported (at least) by motif or binding data.

We have performed suggested motif analysis and changed the language in this section to better fit the data.

3. Replication and dataset comparability. Both ATAC-seq and soxB1-2 RNA-seq were performed on head fragments, but the number of replicates differ between assays (ATAC-seq n=2 per group, RNA-seq n=4-6). This is of course acceptable, but when interpreting the results, it should be taken into consideration that the statistical power is different when using data collected using different techniques and having a varied number of replicates.

Thank you for raising this important point regarding replication and comparability across datasets. We agree that the differing number of biological replicates between the ATAC-seq and RNA-seq experiments results in different statistical power across assays. We have now clarified this consideration in the manuscript text.

Minor comments

"Thousands of accessible chromatin sites". Please state the number of peaks and the thresholds for calling them. Ensure consistency between text (264 DA peaks) and Figure 1 legend (269 DA peaks).

We have clarified specific peak numbers and will include the calling parameters in the methods section. Additionally, we will fix the discrepancies between differential peaks.

Specify the y-axis normalization units in all coverage plots.

We have specified this across plots.

Clarify replicate numbers consistently in the text and figure legends.

We have identified and corrected discrepancies in the figure legends vs text and correct them and ensured they are included consistently across datasets.

Referees cross commenting

The reviews are highly consistent. They recognize the value of the work, and raise similar points. The main shared view is that the current data do not distinguish direct from indirect effects, and claims about pioneer activity should be softened, and further analysis of the differentially accessible peaks could strengthen the link between SoxB1-2 and the chromatin changes.

- I don't think that it's necessary to further characterize experimentally mecom or castor (as suggested), but of course that it could have value.

We thank all three reviewers for their positive assessment of the value of our work aiming to elucidate mechanisms by which SoxB1-2 programs planarian stem cells. In the revision,

we have improved the presentation and carefully edited conclusions about the function of SoxB1-2. Performing motif analysis and GO annotation of upregulated genes has strengthened our observation that SoxB1-2 acts as an activator and has revealed putative binding sites.

The preliminary revision does not yet include further characterization of *mecom* and *castor* downstream genes. In response to Reviewer #2, we appreciate that additional validation of the *mecom* and *castor* RNA-seq results and further phenotypic characterization would strengthen this section. Although we are currently conducting in situ hybridization experiments to validate transcriptional changes in *mecom* and *castor* using the same experimental framework applied to soxB1-2 downstream candidates, we also reconsidered, as we did in our first revision, whether this is necessary or better suited for future investigations.

In the revision, we noted that our Discussion points were not balanced and that we emphasized the *mecom* and *castor* results in a manner that distracted from the major focus of the work, likely contributing to the impression that additional experimental evidence was required. Therefore, we have revised the section accordingly and streamlined the Discussion to avoid repetitive statements and to focus on the insights gained into the mechanism of SoxB1-2 function in planarian neurogenesis. We remain open to including these additional experiments if the reviewers or handling editors consider them essential; however, we agree that their inclusion is not absolutely necessary.

Reviewer #3 (Significance (Required)):

General assessment. The study offers valuable observations by combining chromatin and transcriptional analysis of planarian neural differentiation. The integration with in situ validation convincingly demonstrates effects on neural tissues and provides a solid resource for future functional work. However, mechanistic interpretation remains limited, partly because of technical limitations of the system. The data support an important role for soxB1-2 in neural and epidermal lineage regulation, but not direct binding or chromatin-opening activity. The authors have previously published analysis of soxB1-2 in planarians, so the addition of ATAC-seq data contributes to solving another piece of the puzzle.

Advance.

This is one of the first studies to couple ATAC-seq and RNA-seq in planarian tissue to dissect regulatory logic during regeneration. It identifies new candidate regulators of sensory and epidermal differentiation and identifies soxB1-2 as a likely upstream factor in ectodermal lineage networks. The work extends previous studies on soxB1-2 activity and neural cell production by integrating chromatin and transcriptional layers. In that respect the results are very solid, although the study remains correlative at the mechanistic level.

Audience.

This work will potentially interest researchers interested in regeneration and transcriptional networks. The datasets and gene lists will be valuable references for follow-up studies on planarian ectodermal lineages, and therefore will appeal to this community.

December 18, 2025
RE: GENETICS-2025-308887

Dr. Ricardo M. Zayas
San Diego State University
Department of Biology
5500 Campanile Dr
San Diego, California 92182-4614

Dear Ricardo:

Congratulations, your manuscript titled "SoxB1-Mediated Chromatin Remodeling Promotes Sensory Neuron Differentiation in Planarians" is accepted for publication in GENETICS! Many thanks for submitting your research to the journal.

We include a few suggestions for improving the manuscript that you may want to consider. You can view these comments at the bottom of this email.

To Proceed to Publication:

1. Format your article according to GENETICS style: <https://academic.oup.com/genetics/pages/author-guidelines>
2. Ensure that you comply with data and community resource citation guidelines:
<https://academic.oup.com/genetics/pages/author-guidelines#section-5-9-2>
3. Upload your final files at <https://genetics.msubmit.net>
4. Add oupsupport@scipris.com and genetics.oup@novatechset.com (or the domains @scipris.com and @novatechset.com) to your email program's "safe senders" list. You will be contacted by both at various points during the production process.

Notes:

- Your currently-accepted manuscript (unedited, as submitted, reviewed, and accepted) will be published at GENETICS and deposited into PubMed as an Advance Access article. Notify sourcefiles@thegsajournals.org before signing your license if you do not wish to publish your article via Advance Access.
- We invite you to submit an original color figure related to your paper for consideration as cover art. Please email your submission to the editorial office or upload it with your final files. You can submit a small-sized image for evaluation, and if selected, the final image must be a TIFF file 2513px wide by 3263px high (8.375 by 10.875 inches; resolution of 600ppi). Please avoid graphs and small type.
- After files are sent to Oxford University Press we use SciPris to manage article licensing and payment. If you do not have a SciPris account, you will receive an email from no-reply@scipris.com to sign up to use Oxford University Press' author portal. After logging in, follow the online instructions to sign your license and arrange any payment due.

If you have any questions or encounter any problems while uploading your accepted manuscript files, please email the editorial office at sourcefiles@thegsajournals.org.

Sincerely,

Phillip Newmark
Associate Editor
GENETICS

Approved by:
David Greenstein
Senior Editor
GENETICS

Review comments (if applicable):

In this revised manuscript the authors have satisfactorily addressed the constructive criticisms of the reviewers. Additional suggestions for the authors to consider:

1. The penultimate paragraph of the Introduction breaks the logical flow of this section, discussing the wider impact of applying ATAC-seq to investigate soxB1-2 function before telling readers how the authors are using this approach to study soxB1-2 function. The authors should consider deleting this paragraph, or moving it to the Discussion. Whichever choice they make, because the journal requires a formal Data Availability Statement, which the authors need to add, the final sentence of this paragraph is unnecessary.
2. Page 10: Referring to GO analysis of genes upregulated in soxB1-2(RNAi) animals, the authors conclude, "Rather than indicating a shift toward alternative lineages, including non-ectodermal fates, these signatures are consistent with stress-responsive and homeostatic programs activated following loss of soxB1-2." In the absence of further information, isn't it possible that these genes could simply reflect cell types whose relative abundance changes when soxB1-2 cells are lost? Because the link to homeostatic programs and stress response isn't entirely clear (to me, at least), the authors may wish to shore up their argument or soften their conclusion here.
3. Supp. Figs. 4C and 7 would benefit from headers above each panel to help the reader know what they are looking at.
4. Supp. Fig. 6 can be simplified by showing the soxB1-2 cluster expression a single time; it seems unnecessary to show the same panel four times.

Manuscript number: GENETICS-2025-308887

Point-by-Point Response to Reviewer

Review comments (if applicable):

In this revised manuscript the authors have satisfactorily addressed the constructive criticisms of the reviewers. Additional suggestions for the authors to consider:

1. The penultimate paragraph of the Introduction breaks the logical flow of this section, discussing the wider impact of applying ATAC-seq to investigate soxB1-2 function before telling readers how the authors are using this approach to study soxB1-2 function. The authors should consider deleting this paragraph, or moving it to the Discussion. Whichever choice they make, because the journal requires a formal Data Availability Statement, which the authors need to add, the final sentence of this paragraph is unnecessary.

We have moved the penultimate paragraph of the Introduction to the Discussion, as suggested.

2. Page 10: Referring to GO analysis of genes upregulated in soxB1-2(RNAi) animals, the authors conclude, "Rather than indicating a shift toward alternative lineages, including non-ectodermal fates, these signatures are consistent with stress-responsive and homeostatic programs activated following loss of soxB1-2." In the absence of further information, isn't it possible that these genes could simply reflect cell types whose relative abundance changes when soxB1-2 cells are lost? Because the link to homeostatic programs and stress response isn't entirely clear (to me, at least), the authors may wish to shore up their argument or soften their conclusion here.

We thank the reviewer for this constructive comment. We agree that the original wording drew conclusions that could not be fully supported by the GO analysis alone. We have revised this paragraph to soften the interpretation, clarify alternative explanations, and more precisely reflect the limitations of the analysis.

3. Supp. Figs. 4C and 7 would benefit from headers above each panel to help the reader know what they are looking at.

We have added headers to the figures to improve clarity and readability.

4. Supp. Fig. 6 can be simplified by showing the soxB1-2 cluster expression a single time; it seems unnecessary to show the same panel four times.

We thank the reviewer for identifying this redundancy. We have revised the figure accordingly.